# Prognostic tools and candidate drugs based on plasma proteomics of patients with severe COVID-19 complications

Maryam A. Y. Al-Nesf [1,2,12], Houari B. Abdesselem[3,12], Ilham Bensmail[3], Shahd Ibrahim [1], Walaa A. H. Saeed [1], Sara S. I. Mohammed[1], Almurtada Razok[1], Hashim Alhussain[4], Reham M. A. Aly[4], Muna Al Maslamani[5], Khalid Ouararhni[3], Mohamad Y. Khatib[6], Ali Ait Hssain [1], Ali S. Omrani [5], Saad Al-Kaabi[1], Abdullatif Al Khal[5], Asmaa A. Al-Thani[4], Waseem Samsam[7], Abdulaziz Farooq[8], Jassim Al-Suwaidi[1], Mohammed Al-Maadheed [2,7], Heba H. Al-Siddiqi[3], Alexandra E. Butler [3,11], Julie V. Decock [3,9], Vidya Mohamed-Ali[2,7] & Fares Al-Ejeh [3,9,10 ✉]

COVID-19 complications still present a huge burden on healthcare systems and warrant predictive risk models to triage patients and inform early intervention. Here, we profile 893 plasma proteins from 50 severe and 50 mild-moderate COVID-19 patients, and 50 healthy controls, and show that 375 proteins are differentially expressed in the plasma of severe COVID-19 patients. These differentially expressed plasma proteins are implicated in the pathogenesis of COVID-19 and present targets for candidate drugs to prevent or treat severe complications. Based on the plasma proteomics and clinical lab tests, we also report a 12-plasma protein signature and a model of seven routine clinical tests that validate in an independent cohort as early risk predictors of COVID-19 severity and patient survival. The risk predictors and candidate drugs described in our study can be used and developed for personalized management of SARS-CoV-2 infected patients.

[1] Department of Medicine, Hamad General Hospital, Hamad Medical Corporation, Doha, Qatar. [2] Center of Metabolism and Inflammation, Division of Medicine, Royal Free Campus, University College London, Rowland Hill Road, London NW3 2PF, UK. [3] Qatar Biomedical Research Institute (QBRI), Hamad Bin Khalifa University, Qatar Foundation, Doha, Qatar. [4] Biomedical Research Center, Qatar University, P.O. Box 2713 Doha, Qatar. [5] Communicable Disease Center (CDC) and the Division of Infectious Diseases, Department of Medicine, Hamad Medical Corporation, Doha, Qatar. [6] Hezm Mebairek General Hospital, Hamad Medical Corporation, Doha, Qatar. [7] Anti-Doping Laboratory Qatar, Doha, Qatar. [8] Aspetar Hospital, Orthopaedic and Sports Medicine Hospital, FIFA Medical Centre of Excellence, Doha, Qatar. [9] College of Health and Life Sciences (CHLS), Hamad Bin Khalifa University, Qatar Foundation, Doha, Qatar. [10] Faculty of Medicine, University of Queensland, St Lucia, QLD 4072, Australia. [11]Present address: Royal College of Surgeons of Ireland in Bahrain, PO Box 15503 Adliya, Bahrain. [12]These authors contributed equally: Maryam A.Y. Al-Nesf, Houari B. Abdesselem. ✉email: FAlEjeh@hbku.edu.qa

The rapid and widespread dissemination of the severe acute respiratory syndrome coronavirus 2 (SARS-CoV-2) has pressured healthcare systems globally. To date, there have been over 60 million individuals infected worldwide, leading to over 1.5 million deaths due to severe complications from the Coronavirus disease 2019 (COVID-19). The International Severe Acute Respiratory and Emerging Infections Consortium (ISA-RIC) released its latest comprehensive report on 14 July 2021 including data from 30 January 2020 to 25 May 2021 for 442,643 individuals with laboratory-confirmed SARS-CoV-2 infections from more than 1600 sites across 61 countries. Patients were split equally between males (221,591) and females (220,390), with a median age of 60 years. The most common comorbidities at admission were hypertension (41%), smoking (35%), diabetes mellitus (28%), cardiovascular disease (17%), and obesity (12%)[1]. The five most common symptoms at admission were shortness of breath, cough, history of fever, fatigue, and altered consciousness or confusion. Oxygen saturation (SpO2%) less than 94% was present in 34.8% and 25.3% of the patients who were and were not on oxygen therapy at admission, respectively. Admission to intensive care or high dependency units (ICU/HDU) at some point of illness, which could be defined as severe COVID-19, was reported for 70,476 (15.9%) patients with an estimated case-fatality ratio of 37.9%; the overall estimated case-fatality ratio is 24.9%[1].

Several studies reported symptoms and comorbidities associated with severe COVID-19 complications; however, early prognostic tools to stratify the risk of developing complications are imperative. In this study, we hypothesized that changes in plasma proteins offer prognostic molecular profiles and can help identify the most informative clinical features presented at admission, which can predict the risk of developing complications. To address this, we used proteomic panel-profiling of plasma from patients with severe complications versus mild-moderate symptoms and control subjects to characterize biological processes and pathways associated with disease pathogenesis and severity. Then we evaluated the plasma proteins and associated routine clinical tests in an independent cohort and examined candidate FDA-approved drugs targeting multiple upregulated proteins and based on biological pathways specific for patients with severe complications.

## Results

**Study cohort characteristics.** Characteristics of the study groups, patients (severe and mild-moderate) and healthy controls, are summarized in Table 1. Most infected patients were males ($n = 91$, 91%). The median age [interquartile range (IQR)] of patients with severe COVID-19 disease defined by admission to ICU (47[35–55] years), but not mild-moderate patients, was higher than the control groups (vs. 38[33–42] years, $p < 0.001$). The ethnicity distribution in the severe and mild-moderate groups was not significantly different; however, the control group had a higher percentage of the Indian subcontinent ethnicity ($p = 0.04$). Patients with severe disease had a significantly higher BMI and were either overweight ($n = 25$, 50%) or obese ($n = 18$, 36%) ($p < 0.001$), and had a significantly higher heart rate and lower SpO$_2$ ($p < 0.001$ for both). Moreover, diabetes and hypertension were significantly associated with severe complications in the lungs and kidneys, compared to mild-moderate disease (Supplementary Data 1).

**High differential protein expression in plasma from patients with severe complications.** Plasma from 50 severe and 50 with mild-moderate COVID-19 patients and 50 control subjects were analyzed using ten different Olink panels (Supplementary

Data 2). For one patient, P064, Olink assays failed QC for seven panels; thus, was excluded. The number of differentially expressed proteins (DEPs) from single panels for samples that passed Olink's QC (Supplementary Fig. 1) is summarized in Fig. 1a. Given the characteristics of our cohort, such as over-representation of males in all groups and younger age of controls, the DEP analyses were corrected for interaction between severity and obesity, sex, age, ethnicity, heart rate, and SpO2. Severe disease versus control identified a large numbers of DEPs; more than 40 out of 92 (>43%) per panel across all panels, whereas the number of DEPs was less in mild-moderate disease versus control. Receiver operating characteristic (ROC) curve analyses using the DEPs in each panel, calculated as a single score, found that all panels significantly classified severe cases versus mild-moderate cases and controls; high area under the curve (AUC, $p < 0.01$) (Fig. 1a).

For a comprehensive molecular view, we carried out the analysis on combined data from the ten Olink panels (893 unique proteins) as a single dataset (Supplementary Data 2). Unsupervised hierarchical clustering, before filtering, revealed that the ten panels could differentiate severe from mild-moderate diseases and controls (Fig. 1b). More DEPs were identified when comparing the severe disease to mild-moderate disease or controls than the mild-moderate disease to controls (Fig. 1c and Supplementary Data 3). Additionally, the DEPs in severe disease versus mild-moderate disease and controls were mainly upregulated, whereas DEPs in mild-moderate versus controls groups had an equal up-and-down-regulation distribution (Fig. 1c).

**Functional analysis of the deregulated proteins in plasma of severe COVID-19 patients.** The DEPs in severe versus mild-moderate, severe versus controls, and mild-moderate versus controls were subjected to Kyoto Encyclopedia of Genes and Genomes (KEGG) pathways enrichment analysis. The statistical significance of enriched pathways should be treated cautiously since our proteomic assays were based on enriched panels; however, relative enrichment is warranted. Cytokine–cytokine receptor interactions were enriched gradually from control subjects to mild-moderate and then severe diseases groups (Supplementary Fig. 2a–c). Such gradual enrichments were observed for several pathways, mainly related to immune, inflammation, and infection and the associated cell signaling pathways.

While standard pathway analyses are informative, the function of proteins in the plasma, particularly during pathology, may not be accurately reflected in such analyses. To this end, we focused on the DEPs in the severe COVID-19 group to carry out functional annotation based on information from databases and literature and concerning their role in circulation and pathogenesis. In total, 375 DEPs were identified in plasma from severe versus mild-moderate groups (Supplementary Fig. 2d, Supplementary Data 3), of which 189 (50%) are secreted proteins based on the Human Protein Atlas secretome[2] and 123 (33%) DEPs are released (secreted or shed) proteins according to literature (Supplementary Data 3, Supplementary Notes). There was no literature for the presence of 64 (17%) DEPs in blood; however, 62 of those were also detected in severe COVID-19 patients in the independent, longitudinal cohort of SARS-CoV-2 infected patients from the Massachusetts General Hospital (MGH cohort)[3] (Supplementary Data 3). Among the 375 DEPs, 288 (77%) were classified into 11 functional groups based on the suggested relationship between functional annotation according to databases and literature searching and COVID-19 pathogenesis and immunity. The remaining 88 DEPs consisted of 64 intracellular or membrane proteins with no described function in the blood and 24 secreted proteins with an unclear role in

**Table 1 Characteristics of patients with COVID19 and controls.**

| Variables | Controls (n = 50) | Mild-Moderate (n = 50) | Severe (n = 50) | Total (n = 150) | P-value |
|---|---|---|---|---|---|
| Age (years) | | | | | |
| Mean ± SD | 37.4 ± 7.7 | 40.0 ± 11.9 | 45.9 ± 11.2 | 41.1 ± 0.9 | <0.001 |
| Median [IQR] | 38 [33–42] | 40 [32–51] | 47 [35–55][a] | 40 [34–49] | |
| Sex n (%) | | | | | |
| F | 2 (4.0) | 8 (16.0) | 1 (2.0) | 11 (7.3) | 0.02 |
| M | 48 (96.0) | 42 (84.0) | 49 (98.0) | 139 (92.7) | |
| Ethnicity n (%) | | | | | |
| Indian subcontinent | 43 (86.0) | 30 (60.0) | 33 (66.0) | 106 (70.7) | 0.04 |
| Middle East North Africa | 5 (10.0) | 15 (30.0) | 10 (20.0) | 30 (20.0) | |
| Others | 2 (4.0) | 5 (10.0) | 7 (14.0) | 14 (9.3) | |
| BMI (kg/m$^2$) | | | | | |
| Mean ± SD | 25.4 ± 4.0 | 26.5 ± 3.9 | 29.7 ± 6.1 | 27.2 ± 0.4 | <0.001 |
| Median [IQR] | 24 [23–27] | 26 [23–28] | 28 [26–33][a, b] | 26 [24–29] | |
| Obesity Level n (%) | | | | | |
| Normal (≤25) | 28 (56.0) | 20 (40.0) | 7 (14.0) | 55 (36.7) | <0.001 |
| Overweight (25–30) | 17 (34.0) | 21 (42.0) | 25 (50.0) | 63 (42.0) | |
| Obese (30+) | 5 (10.0) | 9 (18.0) | 18 (36.0) | 32 (21.3) | |
| Heart rate (beats per minute) | | | | | |
| Mean ± SD | 76.5 ± 10.0 | 89.8 ± 16.1 | 102.4 ± 16.3 | 89.3 ± 1.5 | <0.001 |
| Median [IQR] | 78 [70–82] | 86 [78–104][b] | 100 [88–117][a, b] | 86 [78–100] | |
| SpO$_2$ (%) | | | | | |
| Mean ± SD | 98.7 ± 0.8 | 98.2 ± 2.1 | 93.6 ± 6.9 | 79.2 ± 0.8 | <0.001 |
| Median [IQR] | 99 [98–99] | 99 [97–100][b] | 96 [91–97][a, b] | 78 [72–86] | |
| SBP (mmHg) | | | | | |
| Mean ± SD | 122.5 ± 10.3 | 131.1 ± 17.7 | 125.5 ± 17.6 | 126.4 ± 1.3 | 0.02 |
| Median [IQR] | 121 [116–130] | 129 [119–139][a] | 128 [109–137] | 126 [116–135] | |
| DBP (mmHg) | | | | | |
| Mean ± SD | 77.9 ± 7.4 | 80.2 ± 9.1 | 79.5 ± 12.6 | 79.2 ± 0.8 | 0.50 |
| Median [IQR] | 78 [73–81] | 81 [74–88] | 78 [71–86] | 78 [72–86] | |

*BMI* body mass index, *SBP* systolic blood pressure, *DBP* diastolic blood pressure.
[a]Significantly different compared to control subjects.
[b]Significantly different compared to Mild-Moderate subjects.

COVID-19 pathogenesis (for details, refer to Supplementary Data 3, Supplementary Notes).

The 288 functionally annotated DEPs in severe versus mild-moderate cases included cytokines and chemokines (13 DEPs), markers of innate immunity (6 DEPs), and markers of T or NK cells-mediated immunity (6 DEPs), which are presumably initiated in the alveolus and the interstitium in response to infection before reaching to circulation (Fig. 2). A significant number of DEPs in the severe group were related to immune evasion (33 DEPs), including IL10, which may play a pathological role in COVID-19 severity proinflammation and T-cell exhaustion[4]. Another large functional network is connected to T helper (Th) cell dysfunction in the severe group (31 DEPs), including the inflammatory role of IL6 and the highly inflammatory trans-signaling through the soluble IL6 receptor[5] that maintains local Th17 cells[6]. However, agonists of Th1/Th17 responses (labeled pos. in Fig. 2) are countered by a high level of antagonists in the severe group (labeled neg. in Fig. 2), including soluble IL17RA[7,8], and IL17RB[9], which act as decoy receptors to inhibit the functional effect of IL17 secreted by Th17 cells. Other examples of immune evasion and Th cell dysfunction are detailed in the "Discussion" section (Supplementary Data 3 and Supplementary Notes elaborate further on the function of DEPs).

Three additional functional networks were expressed in the severe group; inflammation (24 DEPs), coagulopathy (27 DEPs), and neutrophil activation and NETosis (program for the formation of neutrophil extracellular traps [NETs]) (24 DEPs). Finally, 124 DEPs are related to organ damage; lung damage (14 DEPs), endothelial and cardiovascular damage (66 DEPs), and other or multiple organs (44 DEPs) (Fig. 2). Of the 124 DEPs

related to organ damage, 116 and 115 (93%) were significantly deregulated on day 0 (at admission) and day 3 in patients who were eventually intubated or died (severe group) by day 28 in the MGH cohort[3] (Supplementary Data 3). Overall, 364 (97%) out of the 375 DEPs in severe versus mild-moderate and controls groups were also significantly deregulated in the MGH cohort in patients who eventually developed severe COVID-19 on admission (day 0) and day 3, respectively. This suggests that dysregulation of these proteins is an early event and not restricted to ICU-admitted patients, as in our cohort. Furthermore, 330 proteins were deregulated in the same direction in our cohort and the MGH cohort; 91% concordance (Supplementary Data 3).

Next, we analyzed the interactions between the 288 proteins (Fig. 2) using the STRING database[10] (STRING-db, version: 11.0). We found that 587 interactions with a STRING-db confidence score of 0.7 or higher connected 213 (74%) of these proteins, suggesting a complex biological interplay between the 11 functional networks (Supplementary Fig. 3). Among the clinically available blood biomarkers in our cohort, C-reactive protein (CRP) concentration correlated with most DEPs (308, 82%), followed by creatinine (242, 64%), urea (210, 56%), and glucose levels (194, 52%) (Fig. 3). In respect to blood cells, white blood cells (WBC) counts had the highest frequency of correlations with DEPs (179, 48%), followed by neutrophils (92, 24%), platelets (74, 20%), and eosinophils (33, 9%), whereas lymphocyte and monocyte count correlated only with 20 (5%) and 7 (2%), respectively (Fig. 3). At the individual functional groups level (Fig. 2), CRP, followed by creatinine, showed the highest frequency of correlations with DEPs in all groups. WBC followed by neutrophils showed the highest frequency of correlations with DEPs in all functional groups (Supplementary Fig. 3), but it was

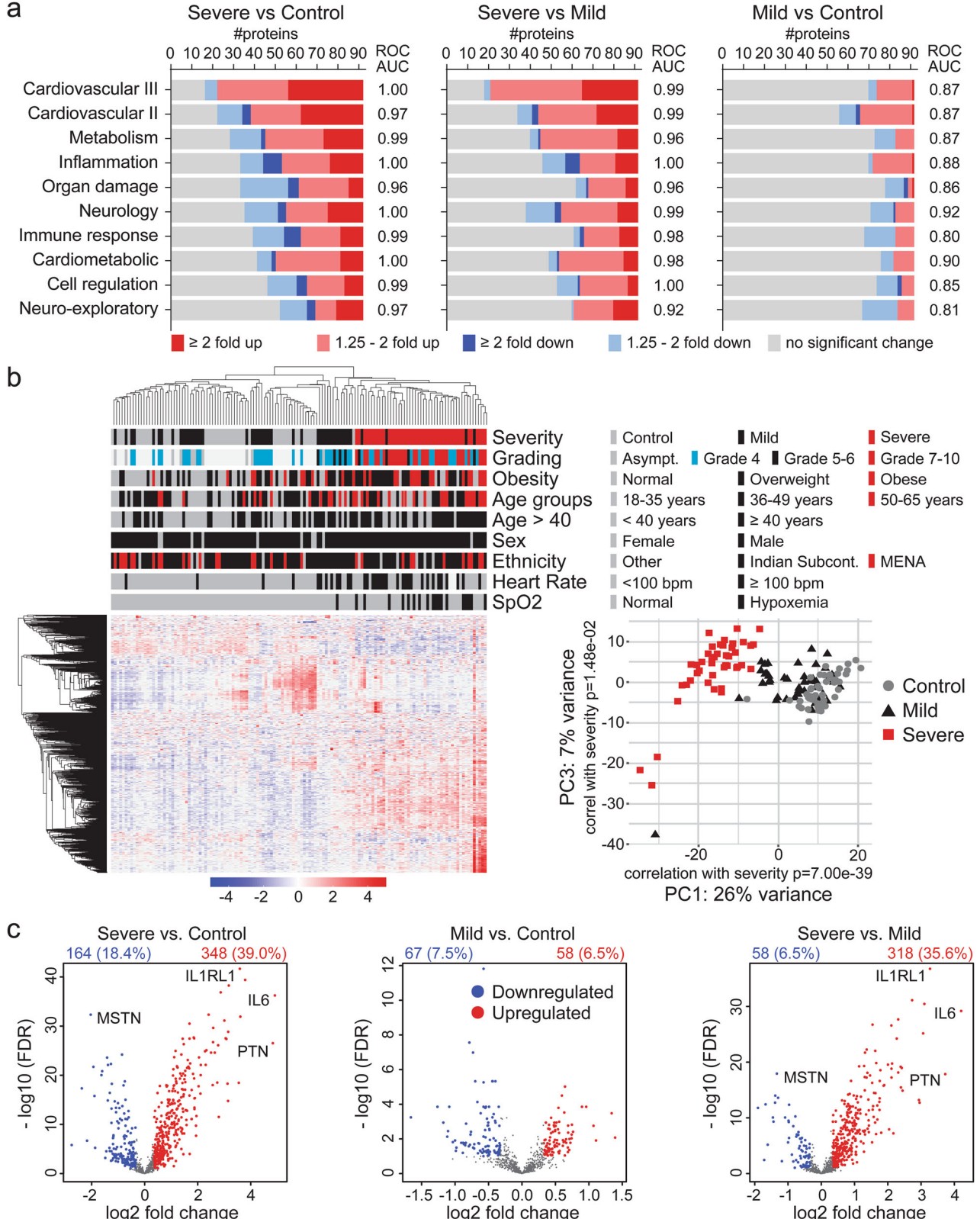

notable that blood cell counts generally had lower frequency of correlations with DEPs compared to biochemical markers.

**Potential drugs to target deregulated proteins in COVID-19 patients with severe complication.** In addition to targeting the enriched KEGG pathways (Supplementary Fig. 2) such as TNFα, coagulation, or JAK-STAT, an analysis of protein-drug interaction

(PDI) was carried out based on the upregulated proteins summarized in Fig. 2 in patients with severe versus mild-moderate disease groups. The Drug-Gene Interaction database[11] (DGIdb, v4.2.0) was screened for FDA-approved drugs which interact with the significantly upregulated proteins (>1.5-fold) in severe group versus both the mild-moderate and control groups. We identified 215 FDA-approved drugs that targeted 74 proteins and were

**Fig. 1 Differential protein expression in plasma from patients with active SARS-CoV-2 infection.** The limma package was used to identify differentially expressed proteins (DEPs) from the single Olink panels and the combined dataset (893 unique proteins), which was defined as protein with more than 1.25-fold change with a $P$-value < 0.05 and FDR < 0.1. **a** Summary of the number of DEPs in each of the ten Olink panels used in the study. DEPs were used to calculate a score for each panel (refer to "Methods"), which was used for ROC curve analysis and the AUC under the ROC curves is stated for each panel. All ROC curves AUC had a $P$-value < 0.01. DeLong et al. method[70]. **b** Unsupervised hierarchical clustering based on all proteins (a total of 893 unique proteins) assayed using the ten Olink panels showed a separation between patients with severe complications compared to mild cases and controls. The heatmap shows z-scores and clustering was done using correlation and average linkage. Principal component analysis (PCA) confirmed the separation of the severe cases based on the expression profiles of all proteins. **c** Volcano plots summarizing the DEPs across the patient groups. Differential expression analysis addressed severity as the main effect and included all factors, from obesity to SpO₂ (except for disease grading), to correct for the interaction of these factors with severity. The time between admission to blood collection was also considered for interaction with disease severity in the comparison between severe and mild cases (right volcano plot in **c**). The number and percentage of the DEPs relevant to all proteins assayed are stated in each panel. Similar analyses were carried out for each panel and shown in Supplementary Fig. 1.

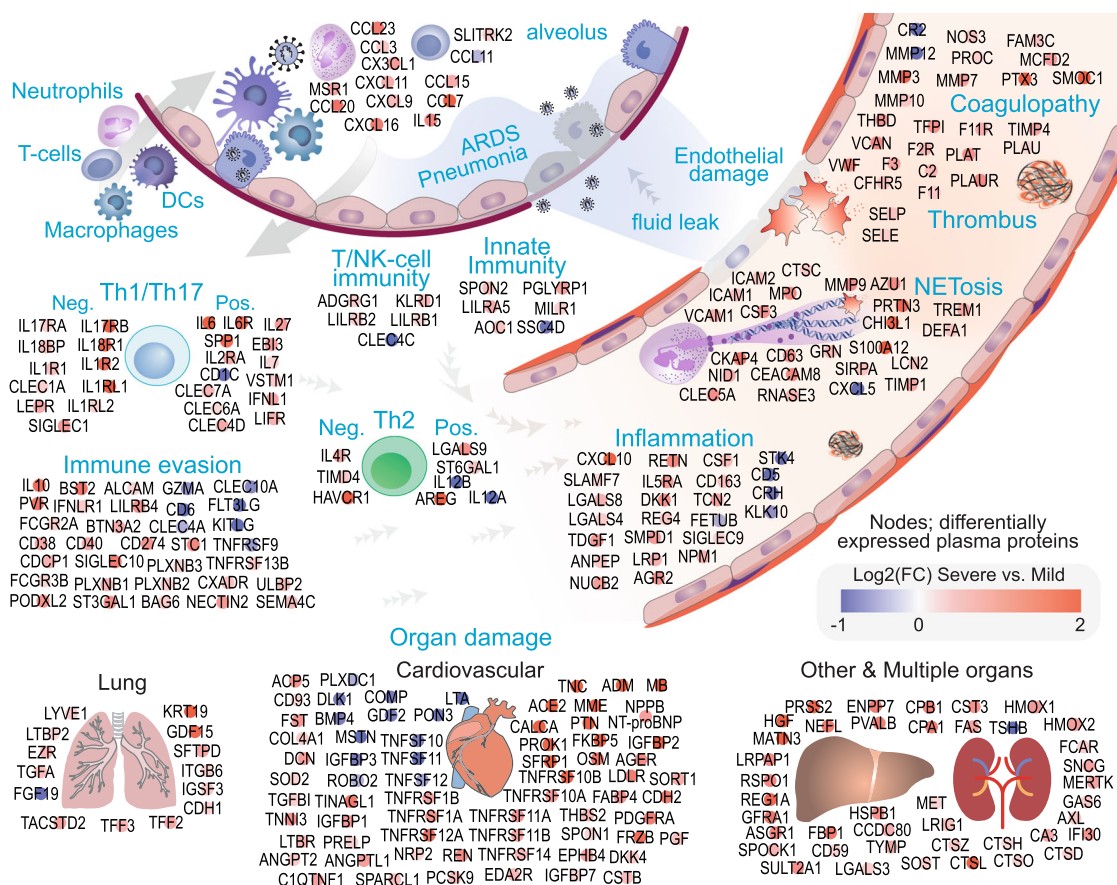

**Fig. 2 Functional analysis of deregulated plasma proteins in severe versus mild COVID-19 disease.** Differentially expressed proteins (DEPs) in patients with severe complications compared to mild-moderate disease were subjected to network analysis using the STRING-db (Supplementary Fig. 3) and annotation for their function as circulating proteins (Supplementary Data 3 and Supplementary Notes). Of the 375 DEPs (1.25-fold change in severe vs. mild cases), 288 (77%) DEPs shown in the Figure could be allocated to 11 functional groups considering their potential function as circulating proteins; chemotaxis, coagulopathy/fibrinolysis, immune evasion, innate immunity, T- or NK-cell immunity, T-/Th-cells dysfunction, inflammation, neutrophils/ neutrophil extracellular traps (NETosis), and organ damage (lung, cardiovascular or other and multiple organs). The remaining 87 DEPs were either known to exist in circulation with unclear function or with known function but with no literature supporting their secretion or release into the blood (see Supplementary Data 3 and Supplementary Notes). The color intensities (red: upregulated, blue: downregulated; legend) depict the log2 fold-change between severe and mild-moderate cases. DEPs are classified as agonists (pos.) or antagonist (neg.) for the Th1/Th17 and Th2 immune responses. Network interactions between the 278 DEPs and their correlation with clinical blood test are shown in Supplementary Fig. 3.

grouped according to drug classes (sheet a of Supplementary Data 4). Briefly, the FDA-approved drugs list included anti-viral drugs (e.g., ribavirin and ritonavir), anticoagulant and thrombolytic drugs (e.g., tenecteplase), corticosteroids and glucocorticoids (e.g., dexamethasone, methylprednisolone and prednisone), nonsteroidal anti-inflammatory (e.g., aspirin and indomethacin), nonsteroidal antiandrogen (e.g., flutamide), immunosuppressive and immunomodulatory drugs (e.g., anakinra, methotrexate, tocilizumab, atezolizumab, and nivolumab).

A larger number of drugs (113 drugs) target single proteins, which were upregulated by greater than or equal to 2-fold in severe versus mild-moderate groups (Fig. 4a), compared to the 38 drugs that target single proteins which were upregulated by 1.5 to 2-fold in severe COVID-19 (Supplementary Fig. 4). A notable feature of the PDI networks is the large interactions between the

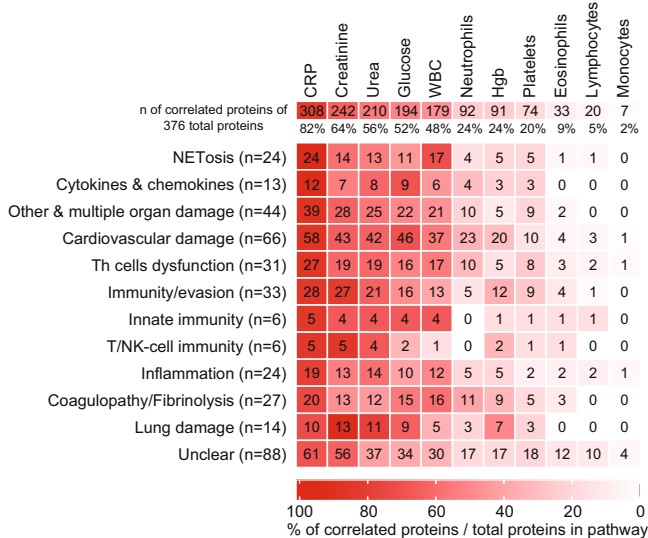

| | CRP | Creatinine | Urea | Glucose | WBC | Neutrophils | Hgb | Platelets | Eosinophils | Lymphocytes | Monocytes |
|---|---|---|---|---|---|---|---|---|---|---|---|
| n of correlated proteins of 376 total proteins | 308 82% | 242 64% | 210 56% | 194 52% | 179 48% | 92 24% | 91 24% | 74 20% | 33 9% | 20 5% | 7 2% |
| NETosis (n=24) | 24 | 14 | 13 | 11 | 17 | 4 | 5 | 5 | 1 | 1 | 0 |
| Cytokines & chemokines (n=13) | 12 | 7 | 8 | 9 | 6 | 4 | 3 | 3 | 0 | 0 | 0 |
| Other & multiple organ damage (n=44) | 39 | 28 | 25 | 22 | 21 | 10 | 5 | 9 | 2 | 0 | 0 |
| Cardiovascular damage (n=66) | 58 | 43 | 42 | 46 | 37 | 23 | 20 | 10 | 4 | 3 | 1 |
| Th cells dysfunction (n=31) | 27 | 19 | 19 | 16 | 17 | 10 | 5 | 8 | 3 | 2 | 1 |
| Immunity/evasion (n=33) | 28 | 27 | 21 | 16 | 13 | 5 | 12 | 9 | 4 | 1 | 0 |
| Innate immunity (n=6) | 5 | 4 | 4 | 4 | 4 | 0 | 1 | 1 | 1 | 1 | 0 |
| T/NK-cell immunity (n=6) | 5 | 5 | 4 | 2 | 1 | 0 | 2 | 1 | 1 | 0 | 0 |
| Inflammation (n=24) | 19 | 13 | 14 | 10 | 12 | 5 | 5 | 2 | 2 | 2 | 1 |
| Coagulopathy/Fibrinolysis (n=27) | 20 | 13 | 12 | 15 | 16 | 11 | 9 | 5 | 3 | 0 | 0 |
| Lung damage (n=14) | 10 | 13 | 11 | 9 | 5 | 3 | 7 | 3 | 0 | 0 | 0 |
| Unclear (n=88) | 61 | 56 | 37 | 34 | 30 | 17 | 17 | 18 | 12 | 10 | 4 |

100    80    60    40    20    0
% of correlated proteins / total proteins in pathway

**Fig. 3 Correlation between the clinical blood markers and the differentially expressed plasma proteins.** The correlation between the expression of the 375 differentially expressed plasma proteins and the available blood markers and blood cell counts in our cohort was evaluated. The overall number (and the percentage) of the DEPs which correlated (significance of $p < 0.05$, two-tailed, using Pearson's correlation, GraphPad Prism) with each clinical measurement are shown in the top row. The heatmap shows the number of DEPs (and the percentage depicted by the heatmap colors), which correlated with each clinical parameter stratified by the functional annotations shown in Fig. 2. CRP showed the highest number of overall and function-specific correlations with the DEPs.

proteins. This may suggest that the benefit from many single-target drugs might be compromised, whereas drugs targeting several proteins may be more promising. To this end, we focused on 66 drugs that target multiple proteins with 2-fold or more (Fig. 4b) or 1.5- to 2-fold (Supplementary Fig. 4) upregulation in severe versus mild-moderate groups. Twenty-five of these 66 drugs target three or more upregulated proteins in severe versus mild-moderate groups (sheet b Supplementary Data 4). The drug atorvastatin targets the highest number of proteins, CXCL10, LDLR, PLAT, TFPI, IL2RA, FAS, and LEPR with 56 interactions. The anti-viral drug ribavirin targets four proteins with more overall interactions (68 interactions); IL6, LDLR, VWF, and CST3. The two forms of a glucocorticoid, prednisone and methylprednisolone, target four proteins with 43 interactions, CALCA, VWF, CXCL10, and IL2RA. The two anti-TNF drugs, etanercept, and infliximab, target the same four proteins (IL6, TNFRSF1A, TNFRSF1B, and KLRD1) with 74 interactions. We deduced drug combinations that would target the highest number of the total 265 interactions (detailed in sheet b of Supplementary Data 4), which included ribavirin + infliximab or etanercept without (7 targets-96 interactions) or with methylprednisolone (10 targets-127 interactions), ribavirin + methylprednisolone without (7 targets-99 interactions) or with cyclosporine (11 targets-167 interactions), and ribavirin + sirolimus without (9 targets-127 interactions) or with methylprednisolone (12 targets-158 interactions). Combining ribavirin and methylprednisolone with sirolimus or cyclosporine targeted the largest fraction (60%) of the 265 interactions between the upregulated proteins in the severe COVID-19 patients.

**The molecular severity score: a 12-protein signature for COVID-19 severity.** To develop blood protein signatures, we used the MUVR tool[12] for variable selection and validation in

multivariate modeling to identify the most stable DEPs that can differentiate all groups' status (severe versus mild-moderate, severe versus controls, and mild-moderate versus controls). We also used Boruta, a wrapper algorithm for all relevant feature selection[13]. By overlapping the features selected by MUVR and Boruta, 35 common proteins were identified (Fig. 5a), of which 12 were selected in 100% of individual 500 variable selection runs (Fig. 5b). These 12 proteins were combined to develop the COVID-19 molecular severity score, which classified severe versus the mild-moderate and controls groups with 100% specificity and 98% sensitivity (AUC under ROC curve 0.999) (Fig. 5c).

We validated the COVID-19 molecular severity score in the independent Massachusetts General Hospital (MGH) cohort[3] (Supplementary Data 5). The score was significantly higher in patients with severe (Acuity 1—death [A1] and Acuity 2—intubated, ventilated but survived 28 days [A2]) compared to non-severe disease (Acuity 3–5, A3–5) (Fig. 6a, b, Supplementary Data 6 for statistical comparisons). The severity scores from plasma collected on days 0 and 3 were highly predictive of COVID-19 severity using the maximum acuity data (A1 or A2 versus the rest, Fig. 6c). The MGH cohort included 57 patients who were intubated at admission but survived for the 28 days, thus, these patients were excluded from the severity analysis to better evaluate the prognostic value of the molecular severity score. The score using day 0 data predicted severity (AUC 0.836) and death (AUC 0.872) between day 3 and day 28 post-admission (Fig. 6d). The score using day 3 data also predicted severity and death (AUC 0.941) between days 7 and 28 post-admission (Fig. 6e). The COVID-19 molecular severity score on day 3 may still be prognostically useful since most of the severity (34 out 52 [60%]) and death (32 out of 42 [76%]) events occurred between days 7 and 28 post-admission. Finally, the molecular severity score in the MGH cohort was significantly higher in severe COVID-19 patients compared to symptomatic SARS-CoV-2 negative patients ($n = 78$) but not different between non-severe virus positive and virus negative patients (Supplementary Fig. 5).

**A molecularly trained clinical score to predict COVID-19 severity.** We hypothesized that the molecular severity score could be used to identify informative clinical parameters and provide scoring system for each parameter to combine them into a single score to predict COVID-19 severity. Eleven out of the 24 clinical parameters available in our cohort were significantly associated with the molecular severity score (Supplementary Fig. 6a), 10 of which significantly classified severe versus mild-moderate disease by ROC curve analysis (Supplementary Fig. 6b). MUVR determined that 7 were most informative (Fig. 7a, Supplementary Fig. 6c, and Supplementary Data 6 for statistical comparisons), which were combined to develop a molecularly trained 7-marker Clinical Score where each clinical measure was weighted according to its molecular severity score. The 7-marker clinical score differentiated severe and mild-moderate groups with 94.7% sensitivity and 85.7% specificity and outperformed each of the single parameters in our cohort (Fig. 7b). Adding the remaining clinical markers (diabetes, SpO$_2$ and/or eosinophils) to the 7-marker clinical score did not improve classification (Supplementary Fig. 6d).

Four out of the 7 markers were also available in the MGH cohort; neutrophil counts, lymphocyte counts, CRP and creatinine levels (Supplementary Data 5). A 4-marker clinical score discriminated severe from mild-moderate groups in our cohort (Fig. 7c), predicted severity in the MGH cohort using data from days 0 (AUC 0.77) and 3 (AUC 0.85), and was more prognostic than the single markers (Fig. 7d). The 4-marker score in the MGH cohort predicted 70% (95% CI 63–77%) and 81% (95% CI 72–90%) risk of severity for

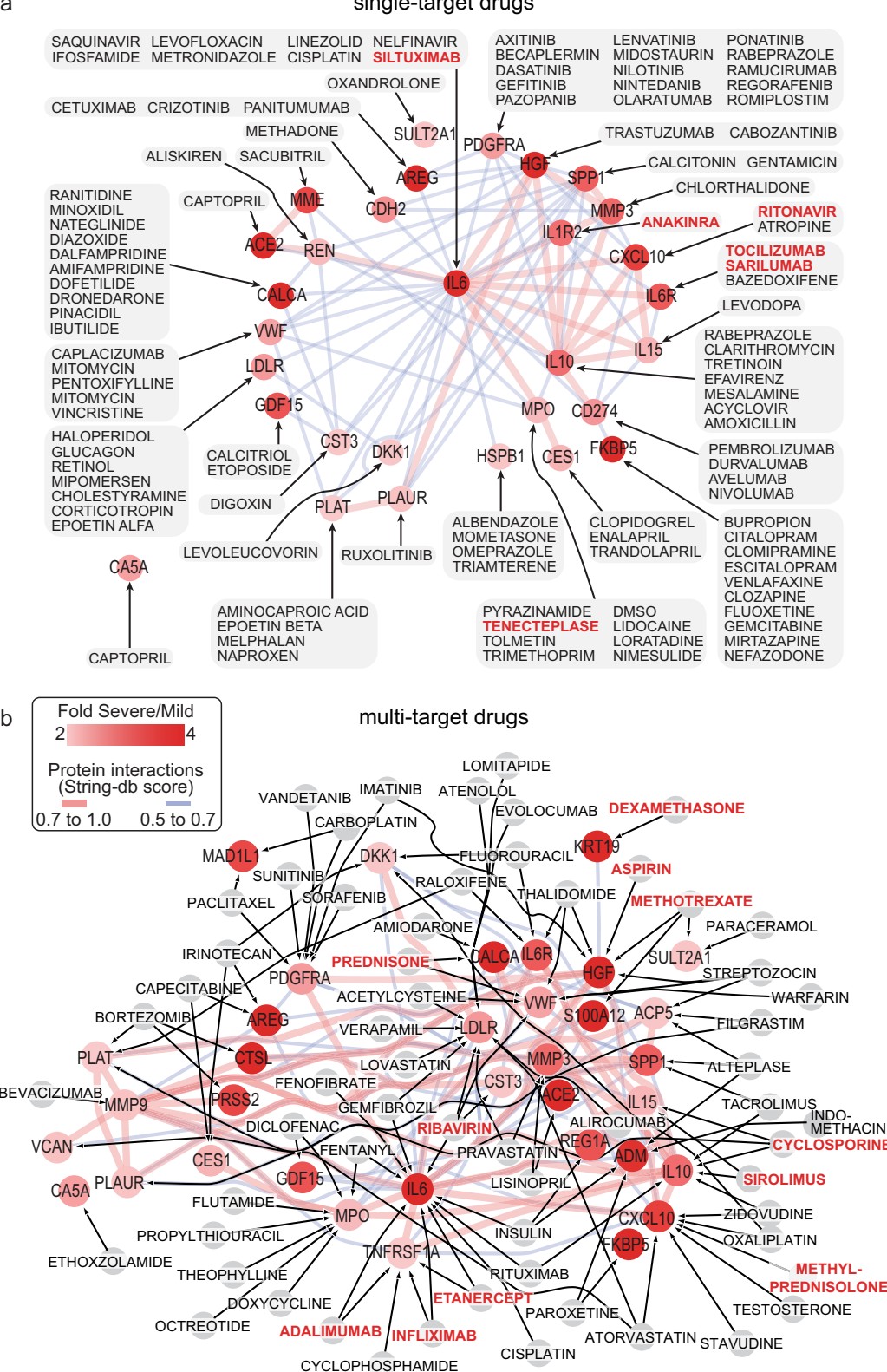

patients with a score higher than 6 compared to 16% (95% CI 12–20%) and 8% (95% CI 4–12%) severity risk for patients with a score less than 4 based on days 0 and 3 data, respectively (Fig. 7d). While the 4-marker clinical score may be more readily available for clinical utility, the molecular severity score was significantly more predictive in the MGH cohort (AUC pairwise comparison; $P = 0.0009$ for day 0 and $P = 0.0004$ for day 3 data). The molecular severity score was more predictive than every clinical feature in the MGH cohort ($P < 0.0001$ from AUC pairwise comparisons), which included age, BMI, pre-existing conditions (kidney, heart, and lung diseases, diabetes, hypertension and immunocompromised conditions), symptoms at presentation (respiratory, fever or gastrointestinal), or blood markers (lymphocytes, monocytes and neutrophils counts, CRP, creatinine, D-dimers, and LDH concentrations).

**Fig. 4 Drug–protein interactions of upregulated plasma proteins in severe COVID-19 patients.** Proteins with more than 2-fold upregulation in severe versus mild-moderate cases were subjected to protein-drug interaction analysis (PDI, using Drug-Gene Interaction database DGIdb, v4.2.0). Target proteins are colored red according to the fold change of expression in severe versus mild cases, whereas drugs are shown in gray boxes or nodes. Drugs that target 1.5- to 2-fold upregulated proteins in severe versus mild cases are shown in Supplementary Fig. 4. Interactions between proteins are depicted by red or blue lines for STRING-db confidence score of 0.7 to 1.0 or 0.5 to 0.7, respectively. **a** Drugs which target single proteins with 2-fold or more upregulation in severe COVID-19 patients versus mild-moderate cases. **b** Drugs which target with two or more upregulated proteins in severe COVID-19 patients. Those multi-target drugs affect proteins shown in (**b**) and/or proteins with 1.5- to 2-fold upregulation in severe versus mild cases (Supplementary Fig. 4).

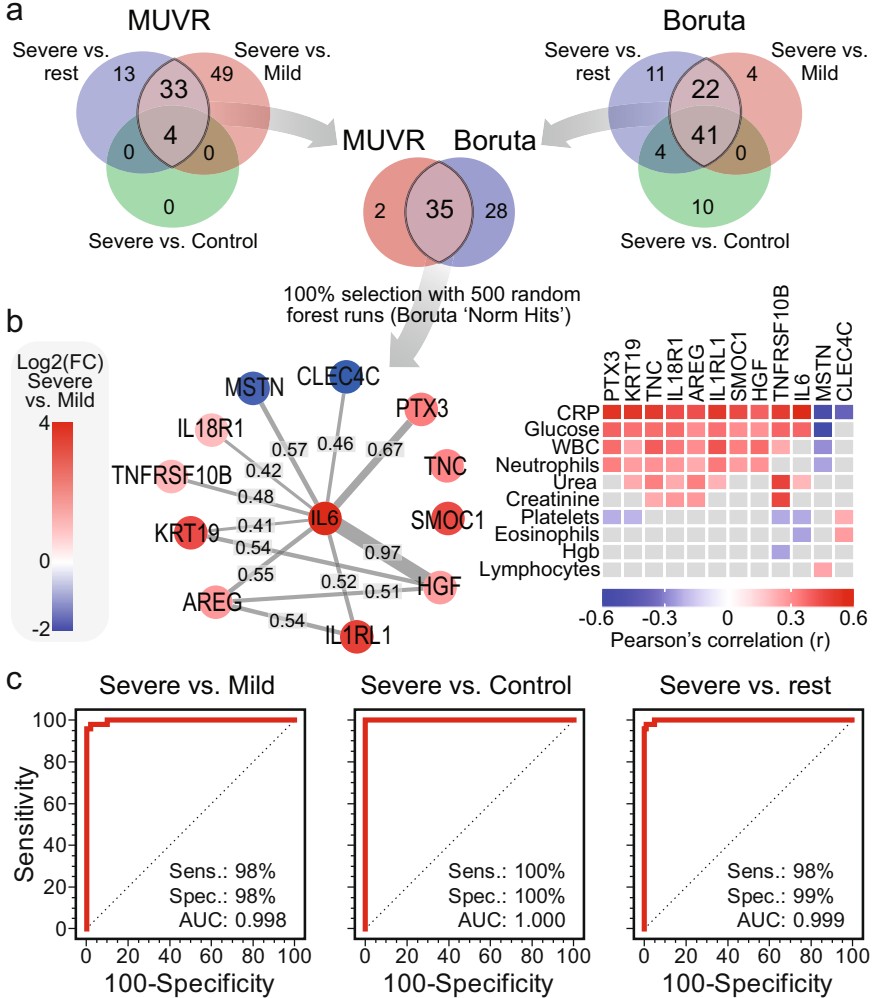

**Fig. 5 A signature of 12 plasma proteins can differentiate COVID-19 cases with severe complications versus mild symptoms. a** Two variable (feature) selection algorithms were used to select the most robust proteins to differentiate severe cases from mild cases and controls; MUVR (multivariate modeling with minimally biased variable selection in R) and Boruta (a wrapper algorithm for all relevant feature selection and feature importance with random selection runs). Proteins that were shared in the differentiation between patients with severe COVID-19 from the rest of the cohort, and specifically from mild-moderate cases, using MUVR and Boruta were overlapped to select 35 proteins, of which 12 proteins were selected at 100% from 500 random forest runs (Boruta 'Norm Hits'). **b** Network analysis for the 12 selected proteins showing the STRING-db confidence score. The heatmap summarizes the significant Pearson's correlation coefficients between the 12 selected proteins and clinical blood markers and blood cell counts. **c** ROC curves based on the 12 DEPs, which were used to calculate the COVID-19 molecular severity score to evaluate the sensitivity, specificity, and the area under the ROC curves (AUC) for differentiating severe COVID-19 cases from mild cases, controls, or both. All ROC curve analyses were significant ($p < 0.0001$ from AUC of 0.5, DeLong et al. method[70]).

## Discussion

The population of Qatar with SARS-CoV-2 infection during the early first wave of the COVID-19 pandemic is demographically unique[14] (predominantly males of younger age) when compared to other populations such as that described in the ISARIC[1]. The cohort of SARS-CoV-2 infected patients in this study was collected early in the pandemic and in the same time frame as the nationwide cohort of the first consecutive 5000 patients with COVID-19 in Qatar[14], thus had similar characteristics of most of the SARS-CoV-2 infected patients (88.7%) being males, with a median age of 35 and the majority (65%) under 45 years old[14]. Between March and April 2020, hospitalizations were predominately under 65 years of age; 95.7% for non-ICU and 85.2% for ICU patients. ICU patients in Qatar were further enriched for

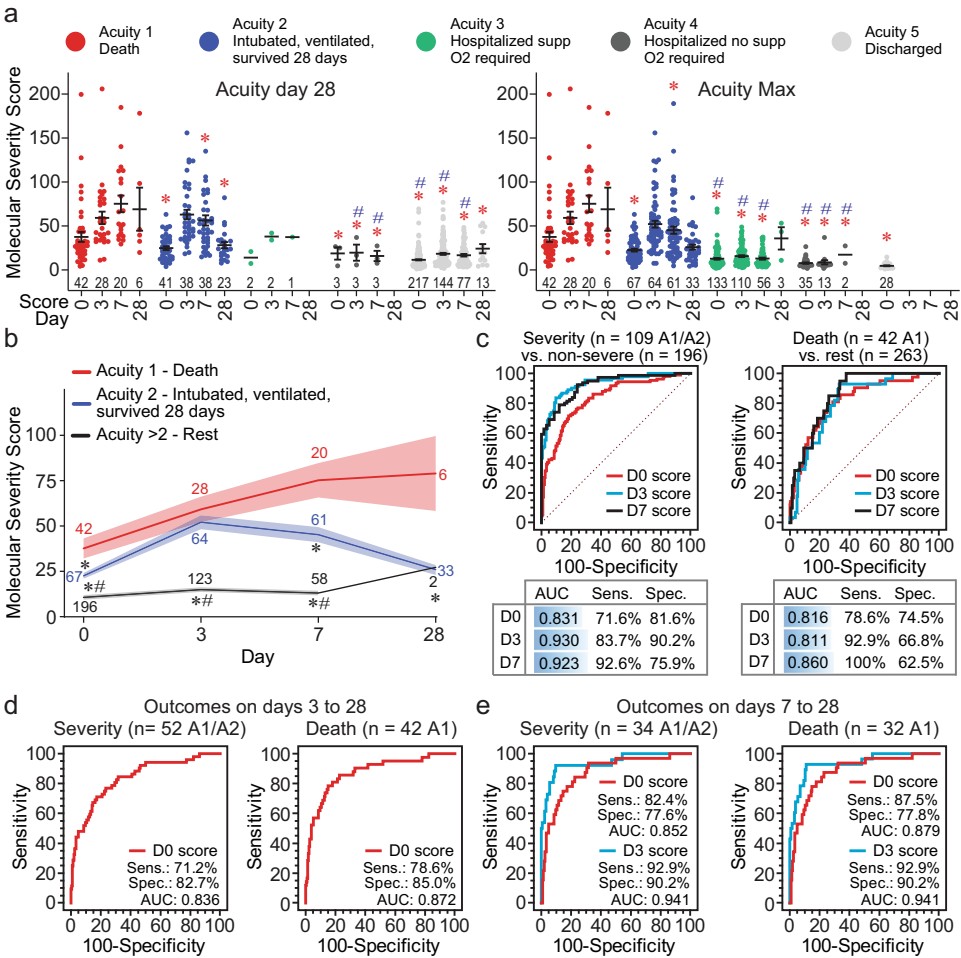

**Fig. 6 Validation of the COVID-19 molecular severity score in the Massachusetts General Hospital (MGH) cohort. a** The molecular severity scores were calculated based on the expression of the 12 proteins measured using the Olink platform in the MGH cohort (Supplementary Data 5). The calculated scores (±SEM) are shown in the scatter plots over time according to the WHO ordinal scale for COVID-19 severity, acuity groups 1–5[3], on day 28 after recruitment (left panel) or the maximum acuity over the 28-day study period (right panel). The number of patients in each group is shown in the bar graph. **b** Time curve of the molecular severity scores (mean ± SEM, number of patients stated for each time point in each group) for the severe COVID-19 groups, A1 group (death) and A2 group (intubated, ventilated but survived 28 days), compared to the remaining groups (A3–A5). * and # in **a**, **b** denote statistical differences ($p < 0.01$, refer to Supplementary Data 6 for exact $p$-values) between the A1 group to the other and the A2 group to the other groups, respectively (two-way ANOVA with Tukey's multiple testing correction). **c** Summary of ROC curve analyses to evaluate the performance of the molecular severity scores on days 0, 3, and 7 in the MGH cohort to predict the maximum COVID-19 severity throughout the 28-day-study. **d, e** Summary of ROC curve analyses to evaluate the performance of the molecular severity scores on day 0 or day 3 to predict COVID-19 severity or death between days 3 to 28 or days 7 to 28, respectively. The AUC, sensitivity (sens.), specificity (spec.), and the number of severe events in each ROC curve are stated. All ROC curves were statistically significant ($p < 0.0001$ from AUC of 0.5, DeLong et al. method[70]).

males (92.6%), and only 16 patients (14.8%) in the ICU were above 65 years or older[14]. Our study selection for patients aged between 18 and 65 and being mainly males with only 9% females was based on the local COVID-19 demographics during the early first wave of the pandemic.

Various differentially expressed plasma proteins were identified in severe COVID-19 patients compared to mild-moderate disease and controls in our study. Typical KEGG pathway enrichment analysis found relative enrichment of pathways in severe versus mild-moderate COVID-19 disease, which have been previously reported in COVID-19. These included cytokine–cytokine receptor interactions and viral interaction cytokine/cytokine receptor interactions related to the cytokine storm[15–17], immune, inflammation, and infection pathways such as TNF and JAK-STAT signaling pathways[15,17], and complement and coagulation cascades[15,18,19].

Patients with severe disease in our cohort had lower lymphocyte counts and the COVID-19 molecular severity score significantly associated with lymphopenia, a prominent feature of

SARS-CoV-2 infection and disease severity[20]. Another feature of severe COVID-19 disease is apparent immunity, particularly T cell responses[20]. While our study did not directly investigate immune cells, the detailed analysis of the 375 dysregulation plasma proteins, particularly for functional role in circulation, shed light on the pathogenesis of severe COVID-19 disease. This functional annotation outlined 11 functional networks including cytokines and chemokines and markers of innate and T and NK cell-mediated immunity which were counteracted by larger networks of immune evasion and T helper (Th) cell dysfunction. For example, severe COVID-19 patients had higher levels of the soluble form of ULBP2, a ligand for the NKG2D receptor on NK cells that mediates mediating cytotoxicity, which inhibits NK-cells as a mechanism to evade immunosurveillance by NK cells[21,22]. Other examples include circulating PD-L1, which induces immune suppression and damage, and associates with COVID-19 pathogenesis and mortality[23], soluble LILRB4 (sLILRB4), which can be produced by a splice variant[24] and suppresses T cell

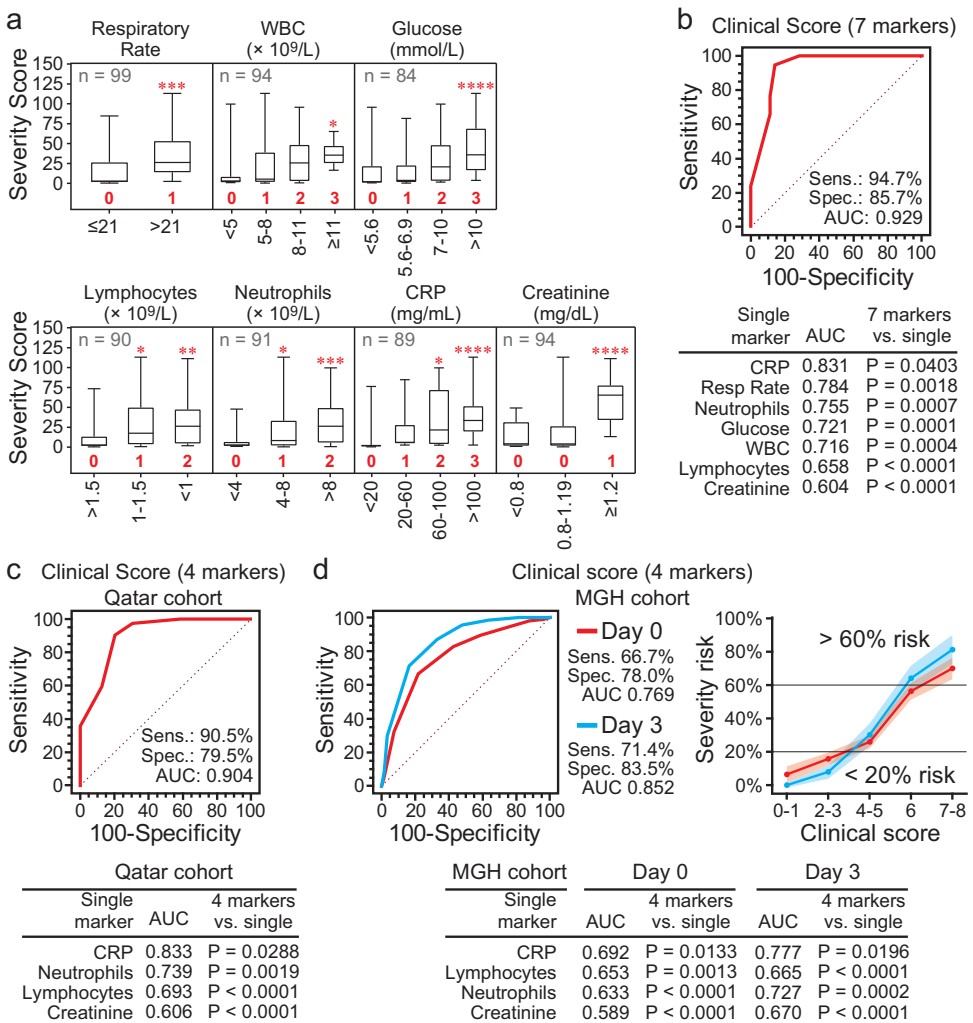

**Fig. 7 A clinical risk score for COVID-19 complications based on the 12-protein molecular severity score. a** The clinical parameters available in the cohort were evaluated for their association with the 12-protein molecular severity score to identify significantly associated parameters and allow scoring (weighting) for the different groups in each clinical measurement. The box plots (median as center line, box marks the 25th and 75th percentiles, and whiskers define minimum and maximum) show the molecular severity score across the groups in the most associated clinical parameters; the number of patients (total of 100, 50 severe, and 50 non-severe) with data available for each parameter is stated in each panel. One-way ANOVA with Dunnett's multiple testing correction was used for clinical parameters with more than two groups, and unpaired two-tailed $t$-test was used for parameters with two groups. Refer to Supplementary Data 6 for more details of the statistical comparisons and exact $p$-value. The groups in each of the seven selected clinical parameter (markers) were given a numeric, integer value from 0 to 3 (shown in red bold font) according to the 12-protein severity score. These values were then used to calculate the Clinical Risk Score by adding the values across the 7 markers for each patient. Refer to Supplementary Fig. 6 for details of variable selection. **b** ROC curve analysis confirmed the significant predictive value of the Clinical Risk Score, which combined the 7 clinical markers. **c, d** Of the 7 markers in (**a**), 4 (CRP and creatinine levels and lymphocyte and neutrophil cell counts) were available in the MGH cohort, thus, were used for independent validation (Supplementary Data 5). **c, d** Show the ROC curve of the Clinical Risk Score based on the 4 markers in our cohort from Qatar and the MGH cohort (from day 0 and day 3 data) from the US, respectively. D Also shows the risk of COVID-19 severity (% risk with 95% confidence interval) in the MGH cohort according to the 4-marker Clinical Risk Score. For **b–d** the Clinical Risk Scores outperformed each of the single clinical parameters in pairwise comparisons ($p < 0.0001$, DeLong et al. method[70]).

responses and elicits T cell anergy or activation of Treg or T suppressor cells[24–26], and PVR (CD155) which is a ligand for CD226 (DNAM-1) and TIGIT expressed on NK cells and a subset of T cells[27], but the soluble form inhibits NK cells CD226-mediated cytokine production[28]. NECTEN2 (also called PVRL2), a ligand for DNAM-1 and PVRIG on NK and T cells[27], has a soluble form (sNECTIN2) which is inhibitory of function[29,30] and was elevated in plasma of severe COVID-19 patients in our cohort. Both TIM-1 (HAVCR1) and its ligand TIM-4 (TIMD4) were upregulated in the plasma of severe COVID-19 patients. The soluble form of TIM-1 may be inhibitory of cellular function[31] including its role in regulating Th2 responses[32]. Likewise, the

soluble form of TIM-4 may be inhibitory of cellular function TIM-4 in Th2 development[31]. Another duo of a ligand and its receptor that was upregulated in plasma of severe COVID-19 is IFNL1 (INF lambda 1, type-III INF) and one of its receptors, IFNLR1. IFNL1 is released by epithelial tissues to bind to IL10RB/IFNLR1 dimers and is involved in antiviral host defense and inhibits Th2 polarization towards Th1[33–35]. However, soluble IFNLR1 (sIFNLR1/sIFN-λR1) inhibits the antiviral and immune effects of type III INF signaling and the induction of IFN-stimulated genes[36]. Further indicators of immune evasion include ST3GAL1 which is carried in circulation by platelets and released upon activation[37] and ST3GAL1-mediated O-linked sialylation of

CD55 act as CD55-mediated immune evasion[38] and the dramatic early proinflammatory IL10 elevation which may play a pathological role in COVID-19 severity proinflammation and T-cell exhaustion[4]. Finally, the T and NK cell-specific serine protease granzyme A (GZMA) for lysis of target cells was specifically downregulated in plasma from severe versus mild-moderate COVID-19 patients in our cohort in line with previously reported reduced levels in COVID-19 severe patients associate with impaired NK- and cytotoxic T cell functions[39,40].

Increased neutrophil counts (neutrophilia)[41,42] and eosinophils counts[42] are known features of severe COVID-19 and were replicated in our study. In contrast CD4[+] and CD8[+] T cells are significantly reduced in severe COVID-19[41,42] but these were not investigated in our study. Nonetheless, a large functional network deduced from the dysregulated plasma proteins in our study pointed to Th cell dysfunction, which is an emerging area in severe COVID-19 immunopathology[17,20,43,44]. Both agonist and antagonist plasma proteins of Th1, Th2, and Th17 were identified. For example, the highly inflammatory IL6 trans-signaling through the soluble IL6 receptor[5] which maintains local Th17 cells[6] is challenged by the upregulated levels of IL27 that potentiates the early phase of Th1 response and suppresses Th2 and Th17 differentiation[45]. Other agonists of Th1 and Th17 responses including soluble IL17RA[7,8], and IL17RB[9] which act as decoy receptors to inhibit the functional effect of IL17 secreted by Th17 cells. Other examples of upregulated soluble, decoy receptors which may skew Th cell responses include IL1R1 and IL1R2 (inhibit IL1B[46]), IL1RL1 (inhibits IL33[47]), IL1RL2 (inhibits IL36[48]), and IL18R1 and IL18BP (inhibit IL18[49–51]). Other functional networks identified in the severe COVID-19 group in our cohort have been previously characterized, including inflammation, coagulopathy, neutrophil activation and NETosis, and endothelial damage[52].

In addition to the high concordance with the MGH study, the plasma proteins identified in our study also confirmed the findings from several studies that identified deregulated plasma proteins associated with COVID-19 severity, ICU admission and mortality[53,54]. Importantly, one study reported a unique neutrophil activation signature composed of neutrophil activators (G-CSF, IL8) and effectors (RETN, LCN2, and HGF), with a strong predictive value to identify critically ill patients whereby the effector proteins strongly correlated with absolute neutrophil count[53]. Our study not only identified those components of the neutrophil activation signature but also found that the COVID-19 molecular severity score also correlated with absolute neutrophil counts. Besides Olink technology, mass spectroscopy has also been used to identify deregulated proteins in sera from SARS-CoV-2 infected patients (e.g., ref. [55,56]). In addition to pathway analysis, these two examples of mass spectroscopy-based proteomics developed predictors of COVID-19 severity. Although none of their serum biomarkers were identified in our study, the biological functions reported in these studies were also captured in our analysis, including complement factors and the coagulation system, inflammation modulators, and pro-inflammatory factors upstream and downstream of IL6. We cannot exclude that the mass spectroscopy-based studies are more comprehensive and less biased than the panel profiling used in our study. However, it should be noted that there was a small overlap between all the proteins detected by mass spectroscopy in sera (before statistical analysis). A more comprehensive comparison between the several published serum or plasma proteomics of COVID-19 patients using the Olink platform and different mass spectroscopy methods is warranted but is beyond the scope of our study.

Our study developed a weighted scoring method to use clinical markers available at admission to predict severity. Several large cohort studies have identified clinical features associated with severity, such as certain comorbidities and older age. These associations, or risk factors, require integration in a model (scoring system) to be used as prognostic tools. A large study by the US National COVID Cohort Collaborative (N3C) based on 174,568 adults with SARS-CoV-2 developed a machine learning model to predict clinical severity using 64 inputs available on the first hospital day with an AUC of 0.87[57]. Another recent study compiled prediction models for COVID-19 severity from 41 studies where more than 60% of the models were from China, the remaining were from Europe or the US, and two were multinational[58]. The compiled models included eight which could be evaluated in the study's independent cohort (University of Illinois Hospital [UIH] Cohort, $n = 516$) and found that the AUCs from these 'external' models ranged from 0.69 to 0.89, while their 'internal' models had AUCs of 0.84 for mortality and 0.83 for criticality[58]. Compared to the N3C study (AUC 0.87), the simpler 7- and 4-marker clinical scores developed in our study had AUCs of 0.93 and 0.90, respectively, and the AUCs from the 4-marker clinical score in the MGH cohort was 0.77 and 0.85 on day 0 and day 3 of admission. Similarly, the AUCs in our study were at least comparable to those from the 'external' and 'internal' models from the UIH cohort study.

In terms of the specific clinical variables, the seven markers in our model were ranked high in terms of importance in the 64-input model from the N3C study (importance ranks using random forest and XG Boost methods were 4 and 23 for glucose, 5 and 3 for respiratory rate, 7 and 10 for WBC, 8 and 53 for creatinine, 11 and 27 for the neutrophil count, 14 and 20 for lymphocyte count, and 16 and 14 for CRP)[57]. The UIH internal models included all the markers of the 4-marker clinical score (CRP, creatinine, neutrophils, lymphocytes) and 5 out of the 7-marker clinical score in our study (WBC in addition to the 4 markers). Interestingly, the UIH study noted that the features used in the tested models were "surprisingly diverse" and the number of variables in each model ranged from 2 through 11[58]. However, the study also noted that three external models (from China) performed well in their cohort from the US, demonstrating that prediction is possible despite geographical and ethnic differences and variations in health systems and during different times of the pandemic[58]. Our study agrees with this conclusion since the 4-marker clinical score from the patients in Qatar performed well in the US MGH cohort. Importantly, the 12-protein COVID-19 molecular severity score reported here was cross-validated in an independent, ethnically different, larger cohort from the Massachusetts General Hospital.

In addition to their potential biomarker value, proteomic profiles can also be used to predict potential drugs for intervention. Our drug–protein interaction analyses shortlisted several FDA-approved drugs that can target the upregulated proteins in severe COVID-19 cases. The classes of potential drugs identified in our analysis is in line with those summarized in the living systematic review and metanalysis of drug treatments for COVID-19[59]. These include the glucocorticoids methylprednisolone and dexamethasone which have been shown to reduce the risk of mortality, mechanical ventilation requirement and length of hospital stay[59]. Other examples identified in our analysis include tocilizumab and sarilumab which target the IL6R and found to reduce the need for mechanical ventilation and the length of hospital stay[59], the anti-IL6 antibody siltuximab which has been recently shown to improve survival in hospitalized COVID-19 patients[60], and the JAK inhibitor ruxolitinib which has been shown to reduce the risk of mechanical ventilation and its duration[59]. Infliximab and adalimumab were two examples of anti-TNF targets identified in our study; an approach suggested for treating COVID-19[61] and are still until trials in the UK (AVID-CC trial[62]) and the US (ACTIV-1 IM trial,

NCT04593940). While aspirin was proposed in our analysis, its effect against COVID-19 mortality is supported in a metaanalysis[63], but only associated with a small increase in the rate of being discharged alive[64]. Guided by drug–protein interactions, our analysis proposed combinations that target the larger number of proteins and interactions where ribavirin with methylprednisolone can be used in combination with infliximab, cyclosporine or sirolimus. Cyclosporine was recently shown to associate with a significant decrease in COVID-19 mortality in a cohort study[65]. Sirolimus (rapamycin) has been proposed to be used against COVID-19 based on preclinical and clinical evidence[66].

In conclusion, our study identified deregulated proteins in the plasma of patients with severe COVID-19 complications that may inform therapeutic interventions. The 12-protein signature identified in our study was developed as the COVID-19 molecular severity score and used to stratify patients according to COVID-19 severity in an independent cohort. The COVID-19 molecular severity score could predict outcomes up to 28 days post-admission and from as early as 3 days of admission. The clinical risk scores, based on 7 or 4 clinical markers, developed in this study uses a simple scoring system of clinical parameters available at the time of admission. The molecular severity and the clinical risk scores developed here have the potential to stratify SARS-CoV-2 infected patients at early stages according to their risk of developing complications to prospectively inform healthcare management and clinical decision-making to prevent complications and mortality.

## Methods

**Patient recruitment.** The study received IRB approval from the Hamad Medical Corporation (HMC, Doha, Qatar) and was supported by a grant from HMC-Medical Research Council (MRC); approval and fund number MRC-05-003. Written informed consent was obtained from all the participants in the study. Participants were not compensated for participation. The conduct of this study was in accordance with the International Council for Harmonization's Guideline for Good Clinical Practice (ICH-GCP) and the Declaration of Helsinki. A cohort of 100 patients (mild-moderate and severe) affected by COVID-19 disease and admitted to Hamad Medical Corporation (HMC) hospitals; tertiary level hospitals in Doha, Qatar, were recruited. Infection was confirmed by positive RT-PCR assays for SARS-CoV-2 from sputum and throat swab with Ct values around 30. Patients with severe COVID-19 were defined as those requiring ICU admissions due to COVID-19 disease or disease complications, while patients with mild-moderate COVID-19 were admitted to community hospitals but did not requiring ICU care. Fifty control subjects were recruited at the Clinical Research Center of the Anti-Doping Laboratory Qatar from volunteers identified by Qatar Red Crescent Society, according to the criteria of being healthy, without prior history of confirmed COVID-19 infection diagnosis, normal SpO₂%, and vital signs. Individuals with poor cognitive ability, or any past or present medical disease or were not able to consent were excluded.

**Samples collection and processing.** Peripheral blood was collected within 5 to 7 days of admission into commercially available EDTA-treated tubes, and plasma and peripheral blood mononuclear cell (PBMC) fractions were separated using Ficoll. Plasma was stored at −80 °C until further analysis.

**Olink proteomic assays.** Plasma samples were profiled in-house using the proximity extension assays (PEA), 96-plex immunoassay developed by Olink Proteomics (Uppsala, Sweden)[67] following the standard protocol at Qatar Biomedical Research Institute's (QBRI) Olink certified proteomics core facility. Quality control and data normalization according to internal and external controls were carried out using the Normalized Protein eXpression (NPX) software, Olink NPX Manager (version 2.1.0.224), and every run was checked and validated by the Olink support team in Uppsala. Ten different panels, each focused on a specific disease or biological process, were used in our study; panel names are stated in the results.

**Bioinformatics.** For the analysis of Olink assays, the protein expression values, as log2 of Normalized Protein eXpression (NPX), were used. Two approaches were used in the analysis; single-panel and combined-panels analyses before confirming the overlap between the two approaches. Olink data that did not pass quality control was excluded from the analyses. R packages for hierarchical clustering (heatmap.2 in version 3.1.1 of the gplots package), principal component analysis

(PCA, prcomp in the Stats R package version 4.1.1), differential expression analysis (Linear Models for Microarray Data, limma version 3.28.14), volcano plots, gene-ontology biological process (GO-PB) and KEGG pathways enrichment analyses were used through the standalone version of iDEP.92 (version .92)[68] installed in RStudio (version 1.2.5). Severe COVID-19 is associated with age and other factors, and infections in Qatar and our cohort had unique characteristics such as enrichment for males and younger age. Thus, the differential expression analyses between the three study groups accounted for interaction between severity as the main effect and other variables, including obesity, sex, age, ethnicity, heart rate, and SpO₂. The median and interquartile range (IQR) for blood collection from severe (5 [IQR 4–7] days) versus mild-moderate groups (4 [IQR 2–6] days) approached significance (t-test p = 0.061). Thus, for more accuracy in the differential expression analysis between severe and mild-moderate groups, we also accounted for the interaction between severity as the main effect and the number of days between admission to blood collection.

For variable selection and validation, we used two algorithms; MUVR[12] (multivariate modeling with minimally biased variable selection in R, version 0.0.975), a statistical validation framework, incorporating a recursive variable selection procedure within a repeated double cross-validation (rdCV) scheme, and Boruta[13] (version 7.0.0), a wrapper algorithm for all relevant feature selection that reports the importance of features and the number of times a feature is selected from repeated runs compared to all other features. Differentially expressed proteins selected by MUVR were used to develop protein signatures represented as meta-protein scores calculated as the ratio of average expression of NPX values of upregulated proteins to the average expression of NPX values of downregulated proteins. Upregulated and downregulated proteins were defined according to the score. For example, if the score was from the comparison of severe versus mild-moderate COVID-19 patients, we used the upregulated or downregulated proteins in the severe versus mild-moderate groups. Scores were evaluated using receiver operating characteristic (ROC) curve analyses to determine the area under the ROC curve (AUC), sensitivity, specificity, and significance (p < 0.05) using MedCalc® (version 12.7, MedCalc Software Ltd., Belgium).

Protein-protein interaction (PPI) was analyzed and visualized using the STRING database (STRING-db version 11.0)[10] accessed through Cytoscape (version 3.7.2)[69]. Protein-drug interaction (PDI) was analyzed using the Drug-Gene Interaction database (DGIdb, version 4.2.0)[11], only using FDA-approved drugs, and interaction networks were visualized in Cytoscape.

**Validation of the COVID-19 molecular severity score in the MGH cohort.** To validate the COVID-19 molecular severity score (the 12-protein signature) developed here, we used the Massachusetts General Hospital (MGH) cohort. The MGH cohort enrolled 384 acutely ill patients, 18 years or older patients, with a clinical concern for COVID-19 upon arrival in the emergency department as described previously[3]. SARS-CoV-2 positivity was reported for 306 patients (80%) while the remaining 78 patients were SARS-CoV-2 negative. The COVID-19 molecular severity score was calculated as described above (meta-protein score) for each patient. The performance of the COVID-19 molecular severity scores in the MGH cohort was evaluated with ROC curve analysis using MedCalc® (version 12.7, MedCalc Software Ltd., Belgium).

Statistics & Reproducibility No statistical method was used to predetermine sample size. No data were excluded from the analyses except for one patient, P064, whose Olink assays failed the internal Olink QC in seven out of the 10 panels used; thus, was excluded in the initial analysis for Fig. 1 and Supplementary Figs. 1 and 2. As per the manufacturer instructions, the plasma samples from the three groups of our cohort were randomized across the Olink plates to minimize bias. The Investigators were not blinded to allocation during experiments and outcome assessment; however, the sample acquisition and processing, Olink data acquisition and internal Olink QC, and the subsequent data analysis and bioinformatics were carried out by independent groups of the Investigators. Patient clinical data analysis was performed using Statistical Package for Social Sciences (SPSS version 26, Chicago IL, USA). Groups were compared using the chi-square test, and Fisher's exact test (two-tailed) replaced the chi-square in the case of a small sample size where the expected frequency is less than 5 in any group. The results were presented as mean ± SD for normally distributed data or median (IQR) for skewed results and/or number and percentage of participants as appropriate. The level of statistical significance was set at p < 0.05. GraphPad Prism (version 8.4.3, GraphPad Software LLC, CA, USA) was used to compare protein signature scores across clinical subgroups using unpaired, two-tailed t-tests or one-way ANOVA with Dunnett's multiple testing correction.

**Reporting summary.** Further information on research design is available in the Nature Research Reporting Summary linked to this article.

## Data availability

All raw data relating to the cohort in this study are supplied in Supplementary Data 2 and were used to generate Fig. 1, Supplementary Fig. 1, Fig. 5c, Fig. 7a–c, and Supplementary Fig. 6. Supplementary Data 3 contains the data used for Fig. 2, Fig. 3, Fig. 5a, b, Supplementary Fig. 2, and Supplementary Fig. 3. Supplementary Data 4 contains the data for Fig. 4 and Supplementary Fig. 4. Supplementary Data 5 includes the

data extracted from the publicly available Massachusetts General Hospital (MGH) cohort published by Filbin et al.[3] and used to generate the graphs in Fig. 6, Supplementary Fig. 5, and Fig. 7d. Supplementary Data 6 details the statistical comparisons related to Fig. 6a, b, Fig. 7a and Supplementary Fig. 6a. The two databases used in this study are available online; String-db (version 11) at https://string-db.org/ and DGIdb (version 4.2.0) at https://www.dgidb.org/.

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

## Acknowledgements

The authors would like to thank all the patients, volunteers, and the healthcare co-workers from Allergy and Immunology Section-HMC, and Dr. Mohamed G.H. Mohamedali, Mr. Hassen Maatoug, and Mr. Ahmed Soliman from Hezm Mebairek General Hospital-HMC for developing disposable racks for samples transportation, tubes labeling, blood collection, and handling. We thank the support provided by Qatar University Biomedical Research Centre, Biosafety Level 3, and Associate Professor Hadi M. Yassine (M.Sc., Ph.D.). We also acknowledge the help of the Anti-Doping Lab-Qatar (ADLQ) and Qatar Red Crescent (QRC) for recruiting control samples. This work was supported by a grant fund from Hamad Medical Corporation (fund number MRC-05-003) and core funding from Qatar Biomedical Research Institute (QBRI).

## Author contributions

Conception: M.AY.A., H.B.A., J.A., M.Al-Maadheed, H.H.A., A.E.B., J.V.D., V.M.-A., and F.A. Designing the study: M.AY.A., H.B.A., M.Al Maslamani, M.Y.K, A. Ait Hssain, A.S.O., S.A., A. Al Khal, A.A.A., J.A., M.Al-Maadheed, H.H.A., A.E.B., J.V.D., V.M-A., and F.A. Conducting experiments and acquiring data: M.A.A., H.B.A., I.B., S.I., W.AH.S., S.SI.M., A.R., H.A., R.MA.A., K.O., and W.S. Analyzing the data: M.AY.A, H.B.A, A.F., and F.A. Writing the manuscript: M.AY.A, H.B.A., H.H.A., A.E.B. J.V.D., V.M-A., and F.A., Revision and editing the manuscript: All authors.

## Competing interests

The authors declare no competing interests.
