## [Peer Review File · Nature Communications]

Prognostic tools and candidate drugs based on plasma proteomics of patients with severe COVID-19 complicationsREVIEWER COMMENTS

Reviewer #1 (Remarks to the Author):

Summary

The authors present a large dataset of O-link proteomic analysis of plasma from N=50 severe, 50 mild, and 50 controls. 'Severe' status was determined by ICU admission. A considerable number of proteins are differentially present between groups, as demonstrated in many different studies. A strength of the present analysis is the depth of the biological pathway analysis, though an immunological interpretation of this data is lacking. Rather, the authors progress to an analysis that seeks to identify drugs that interact with differentially expressed proteins followed by a severity scoring based on these data.

Specific comments

- 1) In the discussion it becomes apparent that only adults aged 18-65 were included in the present study – what is the justification for excluding older patients? The 'Severe' status was determined by ICU admission – no information is available on whether this admission was solely due to respiratory failure, or which WHO respiratory severity categories these 'severe' patients would fall into.
- 2) The cohort is unusually heavily biased towards males, with only 9% of patients being female. Though other studies have shown a male predominance in hospitalised COVID-19, the extent of this in the present study is unusual.
- 3) The methods state that control subjects were age matched to the patients, but this does not appear to be the case. Only summary statistics are given in Table 1, indicating that differences between the 3 groups are present, and medians indicate that the control group are the youngest population.
- 4) A 46 protein molecular severity score is not a clinically practical tool and has not been compared to the numerous other clinically described prognostic and deterioration models available, that are based solely on routine clinical data and are therefore more practical. The subsequent approach of analysis which identifies clinical features most closely aligned with this molecular score is difficult to understand. Far larger cohorts have identified clinical features associated with severity – with much overlap to the present analysis, so what is the benefit of determining the alignment of this with the molecular score, rather than simply associating these with severity? I believe that the greater value here would be in the biological interpretation between linking clinical laboratory features with specific proteomic changes – e.g. linking neutrophilia with neutrophil mediators as currently done, but incompletely explored for other relationships.
- 5) It is hard to understand if patients were already clinically 'severe' at the time of sample collection – the methods state recruitment after 5-7 days of admission. At this time one would expect the 'severe' patients to already be 'severe' – but this cannot be determined from Table 1. The authors assert that the molecular score is predictive for development of severe disease, but the time line of sample collection makes it unlikely that these samples were collected prior to progression to severe disease. If patients were already 'severe' then the data is not of prognostic value, as implied in the results and discussion. The Day 0 MGH data set may be more suitable for such a prognostic effort.
- 6) The main determinant of the clinical risk score is 'Main Diagnosis' which makes it apparent that some patients had ARDS at the time of sampling, while others do not. This further demonstrates that samples were collected at a late time point of disease, when a 'risk' score is not an accurate description of the result generated. Rather, the score in figure 6 appears to simply describe clinical differences between people with severe or mild disease, not factors that indicate a risk/propensity to clinical disease progression. This is not accurately presented in the manuscript.
- 7) In figure 2 DEPs between mild versus severe groups are shown as a network plot. These same DEPs are then taken to figure 3 to look at interactions with drugs. However, the DEPs between figures 2 and 3 appear to be different – for example IL-6 features prominently in Figure 3, but is absent from Figure 2. Why are the DEPs different between figures 2 & 3, when I understand these to discuss the same comparison?
- 8) The drug interactions analysis is interesting and potentially valuable; however, it is surprising that drugs with known efficacy against COVID-19 such as dexamethasone and tocilizumab do not feature in this analysis (though IL-6 signalling is highlighted that relates to the latter). Why do corticosteroids not feature?
- 9) There is a considerable body of literature that has looked at immunological mediators and

pathways in COVID-19, but very little is cited. Much of this literature has performed similar analyses, many with similar results. The authors are encouraged to consider the wider literature in supporting their analyses and interpretation.

Reviewer #2 (Remarks to the Author):

The authors present analysis of panel of proteins linked to COVID-19 disease severity identified in a medium sized cohort of patients using Olink technology. The authors themselves have provided an in-depth comparison of their findings with existing literature and it appears there are many studies which report similar findings which does decrease the novelty of the study somewhat. However, I am in general happy with the quality of the work, and I believe it to be suitable for publication pending some revisions to the manuscript which I detail below.

Page 5

Authors refer to a Chi squared test showing that in severe disease had a skewed distribution toward more upregulated proteins. It is not clear what if any conclusions can be drawn from this. The statement that this “indicates a disruption of biological responses” is such a generic conclusion as to be meaningless. I suggest a revision or clarification of what the authors mean here or remove the statement all together, I don't think it adds anything to the manuscript.

At the end of the paragraph beginning “The DEPs in severe disease” the authors state something about a highly connected network of proteins and then decide to not show the data. Please either include this as supplementary or remove the statement. If it is not important enough to show the data, it can't be essential to the manuscript.

Figure 2 and associated discussion

The authors should reconsider relying on String-db analysis to discuss their findings here. I understand the intention to conduct an unbiased analysis but, it would be more powerful to establish the pathways they want to discuss by citing the literature rather than relying on the “number of connections per protein” as a metric. These “numbers of connections” are also based on an extremely low confidence score cut-off of 0.4. resulting in a greater than 50% FDR. This is not satisfactory to establish a candidate protein-protein interaction, especially as the Olink panels are designed to contain functionally related proteins in the first place.

- Please revise the presentation of figure 2 and the discussion related to it in the manuscript. Do not rely as heavily on String interactions and where String is used, the confidence score cut-off should be increased significantly.

Figure 4 and associated discussion

I have similar issues with panel D of Figure 4. It cannot be claimed that a “single highly interactive network” was identified if the same confidence parameters for this figure were used as in figure 2. I think in this case string could be used to display the network around IL-6, but the confidence score must be dramatically increased. It does not weaken the message to have a higher confidence network and more orphan nodes. It is also not clear what the coloured edges refer to, please update the figure legend to include this information. In my opinion the ROC curves should be the star of the show in figure 4 anyway.

The sentence at the top of page 8 beginning “The complete GO-BP enrichments” is far too broad of a statement to be meaningful and only serves to dilute the message of the manuscript. I would recommend removing it entirely.

- Please revise Figure 4 to include a lower FDR assessment of the regulated network.
- Please revise the discussion of Figure 4 in the manuscript to reflect these changes.

Reviewer #1*Summary*

The authors present a large dataset of O-link proteomic analysis of plasma from N=50 severe, 50 mild, and 50 controls. 'Severe' status was determined by ICU admission. A considerable number of proteins are differentially present between groups, as demonstrated in many different studies. A strength of the present analysis is the depth of the biological pathway analysis, though an immunological interpretation of this data is lacking. Rather, the authors progress to an analysis that seeks to identify drugs that interact with differentially expressed proteins followed by a severity scoring based on these data.

We thank the Reviewer for the summary and comments. Indeed, several studies have reported a considerable number of differentially expressed proteins in COVID-19 patients. We further strengthened our biological pathway analysis to address COVID-19 pathogenesis and immunological interpretations of the data. In addition to protein-protein interactions and standard network analysis, we carried out a careful analysis to deconvolute the function of proteins in the plasma particularly during COVID-19 pathogenesis and immunity. For example, the function of several immune-related receptors, such as IL1R1, IL1RL1, IL1R1, IL17RA and IL17RB, in circulation differ from their role on the cell surface. Although our study did not investigate immune cells, the new analysis of the plasma proteins patterns in the revised manuscript led to a deeper understanding of the disease pathogenesis and immune dysfunction in severe COVID-19.

- **Figure 2 was revised to reflect the function of the plasma proteins in circulation.**
- **Supplementary Table 3 was amended to include annotation to each of the 365 DEPs including cell localization and whether secreted to blood, and functional group classification based on functional description from databases (Human Protein Atlas, UniProt, and NBCI Gene) and literature search.**
- **Supporting File 1 is now included to summarize the functional annotation of the proteins in plasma and includes all cited literature (557 citations) to explain the functional grouping in Figure 2.**
- **Lines 98 to 169 in Results were re-written and new results for the functional network analysis were described.**
- **Lines 257 to 312 in Discussion were added to discuss the results from the new functional network analysis in the context of COVID-19 pathogenesis.**

Specific comments

1) In the discussion it becomes apparent that only adults aged 18-65 were included in the present study – what is the justification for excluding older patients? The 'Severe' status was determined by ICU admission – no information is available on whether this admission was solely due to respiratory failure, or which WHO respiratory severity categories these 'severe' patients would fall into.

This is an important point. The definition of the 'severe' status is addressed in detail in response to the similar comments #5 and #6; briefly, we now provide the WHO respiratory severity categories for the severe patients in our study. We also updated the data from ISARIC both in the introduction and discussion since our initial submission was early during the pandemic in 2020.

- **Lines 44 to 55 in Introduction were re-written to update the ISARIC data**

2) The cohort is unusually heavily biased towards males, with only 9% of patients being female. Though other studies have shown a male predominance in hospitalised COVID-19, the extent of this in the present study is unusual.

We agree that the cohort is different from that expected as per other publications (such as younger age and males), however, the population of Qatar is heavily biased towards males, and this is therefore reflected in relation to COVID-19 too. We had outlined the demographics of COVID-19 in Qatar based on the first 5,000 consecutive cases in Qatar. This has been further clarified in the below edits of the discussion.

- **Lines 246 to 256 in Discussion were re-written to address the characteristics of our cohort**

3) The methods state that control subjects were age matched to the patients, but this does not appear to be the case. Only summary statistics are given in Table 1, indicating that differences between the 3 groups are present, and medians indicate that the control group are the youngest population.

We attempted to recruit and collect blood from age, sex and ethnicity-matched control subjects during the challenging first peak of infections in April 2020 but failed to address this completely. Median [IQR] age reported for controls in Table 1 was 38 [33-42] years, not significantly different from the mild-moderate group (40[32-51] years). However, the Reviewer is correct that the control group was younger than the severe COVID-19 group (47[35-55] *) which was stated in the footnote for Table 1 (*Significantly different than controls). This has also been changed in the Methods section. It should be noted that our analysis had addressed differences in age and other features as stated in the legend of Figure 1 stating “*The differential expression analysis addressed severity as the main effect and included all factors, from obesity to SpO2, to account for the interaction of these factors to severity.*”

To address and clarify these valid points raised by the Reviewer, the method and results section were edited as follows:

- **Line 401 in Methods has been edited to include only matching for sex and ethnicity**
- **Lines 426 to 434 in Methods section detailed the correction for interacting variables**
- **Lines 69 to 71 in Results stated the median age of the cohort groups**
- **Lines 83 to 85 in Results clarified the correction for interacting variables**

4) A 46 protein molecular severity score is not a clinically practical tool and has not been compared to the numerous other clinically described prognostic and deterioration models available, that are based solely on routine clinical data and are therefore more practical. The subsequent approach of analysis which identifies clinical features most closely aligned with this molecular score is difficult to understand. Far larger cohorts have identified clinical features associated with severity – with much overlap to the present analysis, so what is the benefit of determining the alignment of this with the molecular score, rather than simply associating these with severity? I believe that the greater value here would be in the biological interpretation between linking clinical laboratory features with specific proteomic changes – e.g. linking neutrophilia with neutrophil mediators as currently done, but incompletely explored for other relationships.

We agree that a 46-protein signature may not be a clinically practical tool. To minimize the protein set, we subjected the differentially expressed proteins (DEPs) to an additional feature selection algorithm, Boruta. The reduced protein signature is described in the Results. Direct comparison of the molecular severity score with published clinical models for predicting severity is not possible as it requires data from the same cohort for accurate statistical comparison. Nonetheless, we address this point using data from the MGH cohort and discussion of the prognostic clinical models.

The subsequent approach of identifying clinical features which align with the molecular severity score allowed defining the cut-offs and weighting for these clinical features to be used in a single model to generate a single score for each patient to predict severity. The 7-marker clinical score, after revision according to comment #6 from Reviewer 1, discriminated severe COVID-19 cases from mild cases with high sensitivity and specificity in our cohort and the combined score outperformed the single markers. We now include results from the MGH cohort based on Day 0 and Day 3 data where 4 out of the 7 markers from our model were available. We show that the 4-marker model is predictive of severity in the MGH cohort and that the model is more predictive than each marker alone. Finally, regarding the benefit of the molecular severity score versus using predictive clinical features, pairwise comparison of AUCs in the MGH cohort showed that the molecular severity score is significantly more predictive than the 4-marker clinical score ($p = 0.0009$ for Day 0 data, $p = 0.0004$ for Day 3 data) and each of the available clinical characteristics which included age, BMI, pre-existing conditions (kidney, heart and lung diseases, diabetes, hypertension, and immunocompromised conditions), symptoms at presentation (respiratory, fever or

gastrointestinal), and blood markers (absolute lymphocytes, monocytes and neutrophils, CRP, creatinine, D-dimers and LDH).

We appreciate the comment about the great value of biological interpretation between linking clinical laboratory features with specific proteomic changes. Our response to this point was added to the new network and functional analysis which was done in response to comments from Reviewer 2. The revised Figure 2 and Figure S3 now include functional network analysis of the significant DEPs between severe cases and mild cases/controls. For each network we provide a heatmap for the correlation of the proteomic data with the clinical blood counts and clinical markers available in our cohort. We also included a new Figure (Figure 3) that summarizes the extent of significant correlations between the plasma DEPs in each of the functional networks and each of the clinical features.

- **Lines 195 to 218 in Results were re-written to describe the reduced protein signature**
- **Lines 219 to 244 in Results were re-written and we have added new findings for the Clinical Risk Score, and compared the molecular severity score with the clinical risk score in the MGH cohort**
- **Lines 332 to 361 in Discussion were added to address findings from large cohort regarding predictive clinical features**
- **Lines 106 to 160 in Results were re-written to describe new analyses and findings relating to the biological interpretations and linking to clinical laboratory features.**
- **New Figure 3 was included**
- **New annotation of function for the DEPs was added to Supplementary Table 3**

5) It is hard to understand if patients were already clinically 'severe' at the time of sample collection – the methods state recruitment after 5-7 days of admission. At this time one would expect the 'severe' patients to already be 'severe' – but this cannot be determined from Table 1. The authors assert that the molecular score is predictive for development of severe disease, but the time line of sample collection makes it unlikely that these samples were collected prior to progression to severe disease. If patients were already 'severe' then the data is not of prognostic value, as implied in the results and discussion. The Day 0 MGH data set may be more suitable for such a prognostic effort.

6) The main determinant of the clinical risk score is 'Main Diagnosis' which makes it apparent that some patients had ARDS at the time of sampling, while others do not. This further demonstrates that samples were collected at a late time point of disease, when a 'risk' score is not an accurate description of the result generated. Rather, the score in figure 6 appears to simply describe clinical differences between people with severe or mild disease, not factors that indicate a risk/propensity to clinical disease progression. This is not accurately presented in the manuscript.

The Review raised several valid and important points in these two comments which relate to each other. We thank the reviewer for raising these as we believe addressing them led to improving the manuscript.

We agree that the severe patients in our cohort are already severe; they were admitted to the ICU before blood collection. The original manuscript and the revised version first focused on characterizing the differences between plasma proteins levels and potential drug targets (up to and including Figure 4) without any claim of prognostication. We then developed a classifier to differentiate the severe cases from mild cases and controls (Figure 5). The claim of prognostic value of the molecular severity score (Figure 6) and the clinical score clinical risk scores (Figure 7) were made based on testing the classifiers we developed from our cohort in the MGH cohort which collected blood on admission (day 0) and days 3, 7 and 28. Particularly, the MGH data on day 0 and day 3 are most relevant for prognostication. Day 3 after admission is still early enough for prognostication. Using the 'Main Diagnosis' at admission in our clinical score model was an oversight; it is too late as correctly noted by the Reviewer- and was excluded from the analysis.

Importantly, the Reviewer's comment on the time of blood collection alerted us that this should have been considered during the DEPs analysis. We had already adjusted the differential protein analysis when comparing the three groups (severe, mild and control) to available clinical features such as sex, age, obesity, and others

(response to comments #1 and #2). However, we did not consider the time of blood collection for the SARS-CoV-2 infected patients (severe and mild-moderate cases). Although not statistically different, we corrected for the time between admission to blood collection for more accurate analysis. The re-analysis to account for the time between admission and blood collection as a variable reduced the DEPs to 524 (vs. 549 in original manuscript) without a major impact on the results.

- **Line 430 to 434 in Methods were added to describe the correction for the time between admission and blood collection**
- **Lines 219 to 244 in Results were edited and now includes the new clinical risk scores, after removing main diagnosis at admission.**
- **Figure 6 now includes more detailed analysis to illustrate the prognostic value of the molecular severity score in the MGH data.**
- **Figure 7 was edited to describe a 7-marker clinical score, and a 4-marker clinical score which was tested and validated in the MGH cohort using day 0 and day 3 data.**

7) In figure 2 DEPs between mild versus severe groups are shown as a network plot. These same DEPs are then taken to figure 3 to look at interactions with drugs. However, the DEPs between figures 2 and 3 appear to be different – for example IL-6 features prominently in Figure 3, but is absent from Figure 2. Why are the DEPs different between figures 2 & 3, when I understand these to discuss the same comparison?

In the original submission for Figure 2, we took the approach to focus on DEPs in severe cases by excluding those that were also differentially expressed in mild-moderate disease versus controls. While Figure 3 did not apply this criterion. We agree that this has discrepancy (e.g. IL-6 not present in Figure 2) and may cause confusion to readers.

- **Figure 2 was changed to address comments from Reviewer 1 relating to biological interpretation of the data, and comments from Reviewer 2 regarding the STRING-db interaction confidence score**
- **Figure 4 (previous Figure 3) was edited for clarity and better presentation of the results.**

8) The drug interactions analysis is interesting and potentially valuable; however, it is surprising that drugs with known efficacy against COVID-19 such as dexamethasone and tocilizumab do not feature in this analysis (though IL-6 signalling is highlighted that relates to the latter). Why do corticosteroids not feature?

In the original version tocilizumab was shown in Figure 3A as an inhibitor of the IL6R. Tocilizumab was highlighted in the text of the results section describing these findings. We also mentioned that corticosteroids and glucocorticoids (e.g. dexamethasone, methylprednisolone and prednisone) in the same text, Methylprednisolone and prednisone appeared in Figure 3, and dexamethasone was shown in Figure S3B.

The presentation of the drug interactions may have been confusing. We revised the presentation of this data for more clarity and revised the results description based on selecting only the DEPs which were upregulated in severe cases versus both mild cases and controls (more stringent criteria to select top drug candidates). The changes to address these points are summarized below:

- **Figure 4 (previous Figure 3) and the Supplementary Figure were edited for better presentation of the drug-protein interactions.**
- **Supplementary Table 4 was edited to list the drugs, and the new results for drug combinations, after adjusting the selection of the drugs to those that target upregulated proteins in both mild-moderate cases and controls.**
- **Lines 161 to 194 were edited; re-written and additional results were added to address the Reviewer comments and describe the results of the adjusted analysis.**

9) There is a considerable body of literature that has looked at immunological mediators and pathways in COVID-19, but very little is cited. Much of this literature has performed similar analyses, many with similar results. The authors are encouraged to consider the wider literature in supporting their analyses and interpretation.

We agree that literature has expanded since the time of our first submission in Dec 2020. Based on our new analyses the discussion section was re-written to include current understanding of immunological mediators and pathways involved in COVID-19 pathogenesis. The discussion was also re-written to describe the drugs suggested in our study considering the rapid development in COVID-19 treatments.

- **Lines 257 to 312 in Discussion describe our findings in the context of the new knowledge about COVID-19 pathogenesis. We also discussed the findings from our new functional analysis which addressed the function of certain proteins and their receptors in circulation specifically.**
- **Lines 326 to 381 in Discussion describe the drug candidates proposed in our study against COVID-19 in relevance to results from clinical trials or findings from cohort studies.**

Reviewer #2 (Remarks to the Author)

The authors present analysis of panel of proteins linked to COVID-19 disease severity identified in a medium sized cohort of patients using Olink technology. The authors themselves have provided an in-depth comparison of their findings with existing literature and it appears there are many studies which report similar findings which does decrease the novelty of the study somewhat. However, I am in general happy with the quality of the work, and I believe it to be suitable for publication pending some revisions to the manuscript which I detail below.

We thank the Reviewer for this summary and comments. We believe that responding to the comments below and the responses to Reviewer #1 in the revised manuscript has increased the novelty of our study and shed new light on COVID-19 pathogenesis, particularly the biological and immunological interpretation of the dysregulated plasma protein patterns in severe COVID-19 patients.

Page 5

Authors refer to a Chi squared test showing that in severe disease had a skewed distribution toward more upregulated proteins. It is not clear what if any conclusions can be drawn from this. The statement that this “indicates a disruption of biological responses” is such a generic conclusion as to be meaningless. I suggest a revision or clarification of what the authors mean here or remove the statement all together, I don’t think it adds anything to the manuscript.

We thank the reviewer for this comment. It was an observation that severe cases versus controls and more so versus mild-moderate cases were skewed towards upregulated protein. We meant disruptive biological responses in terms of increases in cytokines and inflammation markers and others. However, we agree the statement was vague and does not add value to the manuscript.

- **The statement has been deleted**

At the end of the paragraph beginning “The DEPs in severe disease” the authors state something about a highly connected network of proteins and then decide to not show the data. Please either include this as supplementary or remove the statement. If it is not important enough to show the data, it can’t be essential to the manuscript.

The reviewer is referring to the DEPs which included those common to all comparisons carried out (Figure S2); severe versus controls, severe versus mild-moderate cases, and mild-moderate cases versus controls. The next section of the manuscript dealt with the more important, focused point which are the DEPs in severe versus mild-moderate cases. We agree that the statement about the network connectivity of all DEPs (data not shown) should be removed; showing the data (a large network) does not add value to the manuscript story.

- **The lines were deleted.**

Figure 2 and associated discussion

The authors should reconsider relying on String-db analysis to discuss their findings here. I understand the intention to conduct an unbiased analysis but, it would be more powerful to establish the pathways they want to discuss by citing the literature rather than relying on the “number of connections per protein” as a metric. These “numbers of connections” are also based on an extremely low confidence score cut-off of 0.4. resulting in a greater than 50% FDR. This is not satisfactory to establish a candidate protein-protein interaction, especially as the Olink panels are designed to contain functionally related proteins in the first place.

• Please revise the presentation of figure 2 and the discussion related to it in the manuscript. Do not rely as heavily on String interactions and where String is used, the confidence score cut-off should be increased significantly.

We agree that the String-db confidence score should be increased. However, since the first submission of our manuscript in December 2020, COVID-19 literature and understanding of its pathogenesis has rapidly grown. Thus, we believe that protein-protein interactions and routine network analysis is not a major addition. Importantly, we noticed several cell-surface receptors in our and others' plasma/serum proteomic studies with COVID-19 patients; IL1R1, IL1RL1, IL1R1, IL17RA and IL17RB to name a few. To this end, we carried out a careful analysis to understand the plasma proteins in severe COVID-19 patients particularly in pathogenesis.

- **Figure 2 was revised to reflect the function of the plasma proteins in circulation.**
- **Supplementary Figure 3 shows the interaction between the proteins in Figure 2 but only those with String-db confidence score higher than 0.7.**
- **Supplementary Table 3 was amended to include annotation to each of the 365 DEPs including cell localization and whether secreted to blood, and functional group classification based on functional description from databases (Human Protein Atlas, UniProt, and NCI Gene) and literature search.**
- **Supporting File 1 is now included to summarize the functional annotation of the proteins in plasma and includes all cited literature (557 citations) to explain the functional grouping in Figure 2.**
- **Lines 98 to 169 in Results were re-written and new results for the functional network analysis were described.**
- **Lines 257 to 312 in Discussion were added to discuss the results from the new functional network analysis in the context of COVID-19 pathogenesis.**

Figure 4 and associated discussion

I have similar issues with panel D of Figure 4. It cannot be claimed that a “single highly interactive network” was identified if the same confidence parameters for this figure were used as in figure 2. I think in this case string could be used to display the network around IL-6, but the confidence score must be dramatically increased. It does not weaken the message to have a higher confidence network and more orphan nodes. It is also not clear what the coloured edges refer to, please update the figure legend to include this information. In my opinion the ROC curves should be the star of the show in figure 4 anyway.

The sentence at the top of page 8 beginning “The complete GO-BP enrichments” is far too broad of a statement to be meaningful and only serves to dilute the message of the manuscript. I would recommend removing it entirely.

- *Please revise Figure 4 to include a lower FDR assessment of the regulated network.*
- *Please revise the discussion of Figure 4 in the manuscript to reflect these changes.*

We agree with the reviewer; a classifier does not have to be enriched for interactions and the GO-BP is not informative; particularly with the revised and smaller classifier.

- **Figure 4 (now Figure 5) was changed to describe the smaller (12-protein classifier). IL6 remained a central node in this classifier and the (higher) String-db confidence score are stated in the figure.**
- **The statement of a single highly interactive network was removed from the Results.**
- **The GO-BP enrichment results and related discussion were deleted.**

REVIEWER COMMENTS

Reviewer #1 (Remarks to the Author):

That authors should be congratulated on the many improvements made to this manuscript. While I still have some minor concerns about the analysis, I believe these are relatively trivial in comparison to the rest of the paper and are largely appropriately discussed as potential limitations of the study#. I am therefore happy to support the progress of the paper.

Reviewer #2 (Remarks to the Author):

The Authors have addressed the majority of the issues I had with the original manuscript. I still find the use of these complex network diagrams rather confusing, but I'm willing to accept that this might be down to personal preference. I am happy for the manuscript to be published in it's current form.

Reviewer #1

That authors should be congratulated on the many improvements made to this manuscript. While I still have some minor concerns about the analysis, I believe these are relatively trivial in comparison to the rest of the paper and are largely appropriately discussed as potential limitations of the study#. I am therefore happy to support the progress of the paper.

We thank the Reviewer for recognizing the improvements made to the manuscript which were possible by the comments and critique provided during the review.

We provided the source data of our study, including the source data of the cross-validation cohort, which should not only allow replication of our analysis but also allow other researchers to analyze our data using different approaches. We thank the reviewer for supporting the progress of this paper.

Reviewer #2

The Authors have addressed the majority of the issues I had with the original manuscript. I still find the use of these complex network diagrams rather confusing, but I'm willing to accept that this might be down to personal preference. I am happy for the manuscript to be published in it's current form.

We thank the Reviewer for the comments and critique provided during the review which allowed us to address previous concerns. The network diagrams are large and may be complex as we included all the differentially expressed proteins in patients with severe COVID-19 versus those with mild-moderate disease. This is our attempt to provide a comprehensive view of our data. We provided the complete source data for our study, and summaries of our results in the Supplementary Data files. Our dataset and the publicly available dataset from the MGH cohort which we used for cross-validation would provide two independent cohorts for the research committee to carry out further, and more focused, analyses to characterize and understand COVID-19 pathogenesis with simpler biological and functional networks.

We thank the reviewer for supporting the publication of the manuscript.